# An updated area-source seismogenic model (MA4) for seismic hazard of Italy

Francesco Visini[1], Carlo Meletti[1], Andrea Rovida[2], Vera D'Amico[1], Bruno Pace[3], Silvia Pondrelli[4]

[1]Istituto Nazionale di Geofisica e Vulcanologia, Sezione di Pisa, Pisa, Italy

[2]Istituto Nazionale di Geofisica e Vulcanologia, Sezione di Milano, Milano, Italy

[3]Dipartimento di Ingegneria e Geologia, Università degli Studi di Chieti-Pescara, Chieti, Italy

[4]Istituto Nazionale di Geofisica e Vulcanologia, Sezione di Bologna, Bologna, Italy

*Correspondence to*: Francesco Visini (francesco.visini@ingv.it)

**Abstract.** A new probabilistic seismic hazard model, called MPS19, has been recently proposed for the Italian territory, as a result of the efforts of a large national scientific community. This model is based on 11 groups of earthquake rupture forecast inputs and, particularly, on 5 area-source seismogenic models, including the so-called "MA4" model. Data-driven procedures were followed in MA4 to evaluate seismogenic parameters of each area source, such as upper and lower seismogenic depths, hypocentral depth distributions and nodal planes. In a few cases, expert judgement or ad-hoc assumptions were necessary due to the scarcity of data. MA4 consists of 20 seismicity models that consider epistemic uncertainty in the estimations of the completeness periods of the earthquake catalogue, of maximum magnitude values and of seismicity rates. In particular, 5 approaches were adopted to calculate the rates, in the form of truncated Gutenberg-Richter frequency-magnitude distribution. The first approach estimated seismicity rates using earthquakes located in each area-source, while the other approaches firstly calculated the seismicity rates for groups of areas considered tectonically homogeneous, and successively partitioned in different ways the values to the area forming each group.

The results obtained in terms of seismic hazard estimates highlight that the uncertainty explored by the 20 seismicity models of MA4 is at least of the same order of magnitude as the uncertainty due to alternative ground motion models.

## 1. Introduction

A recent project led by the Seismic Hazard Centre (*Centro di Pericolosità Sismica*, CPS) of the Italian *Istituto Nazionale di Geofisica e Vulcanologia* (INGV) and funded by the Italian Civil Protection Department produced a new time-independent Probabilistic Seismic Hazard Assessment (PSHA) model for Italy, called "*Modello di Pericolosità Sismica 2019 - MPS19*" (Meletti et al., 2021). The model consists of a suite of Earthquake Rupture Forecasts (ERFs) and Ground Motion Models (GMMs), respectively described in Visini et al. (2021) and Lanzano et al. (2020), that are based on updated and new data

acquired in the last decade after the release of the current reference Italian seismic hazard model in 2004-2006 (MPS04, *Modello di Pericolosità Sismica 2004*; Stucchi et al., 2011), which is currently adopted as seismic input in the Italian building code (NTC, 2018).

In particular, MPS19 consists of 564 alternative seismic hazard models (i.e. logic-tree branches) obtained by combining 11 groups of ERFs, each made by a different number of sub-models (for a total of 94 ERFs) to consider the epistemic uncertainty inside each group, with a set of 6 GMMs (3 for active shallow crustal regions, 2 for subduction zones and 1 for volcanic areas). In terms of seismic source typologies, 5 groups of ERFs out of 11 consider area sources, 2 are based on smoothed seismicity calculated on a grid of points, 2 combine faults sources with background seismicity, and 2 derive earthquake rates from geodetic data over a grid of points. The ERFs are based on updated and new historical, geological, geodetic and palaeoseismological data sets collected over the last 15 years, since the realisation of MPS04, for the Italian territory and its conterminous areas.

In this paper we describe one of the 5 area-source ERFs, that is the so-called "MA4" model (i.e., area-source model #4, *modello ad aree #4* in Italian), based on the seismogenic zoning ZS16 (*Zonazione sismogenetica 2016*). ZS16 represents the evolution of previous area-source models proposed in the last 30 years as unique area-source input in seismic hazard assessment performed in Italy; these models are all based on the same seismotectonic approach to seismogenic zoning described by Meletti et al. (2000).

The ZS16 of the MA4 model incorporates a number of different parameters for each defined area source: a) geographical boundaries, b) top and bottom depth of the seismogenic layer, c) hypocentre distribution, and d) style of faulting. For each area source of ZS16, MA4 uses 5 alternative frequency-magnitude distributions, providing the annual rates of all earthquakes with Mw ≥ 4.5, that is the threshold magnitude adopted in MPS19 (Meletti et al., 2021; Visini et al., 2021).

In the following, we first briefly introduce the input data considered for developing the ZS16 and the MA4 model, then describe the methods used to define the geometry of area sources and to estimate, for each of them, top and bottom depth of the seismogenic layer, hypocentres distribution, style of faulting, and annual rates of earthquake occurrence. Finally, seismic hazard estimates computed using the MA4 model are presented and discussed.

## 2. Chronology of seismogenic zonings and PSHA models for Italy

The MA4 ERF is based on the area-source approach for defining seismicity parameters. The choice operated in this work was to update previous zonings designated for previous seismic hazard assessment in Italy. Since 1990, several seismogenic zonings had been released adopting the seismotectonic approach described in Meletti et al. (2000). In the first half of the 1990s, the first zoning adopted in a seismic hazard computation for the whole national territory was ZS4 (*Zonazione sismotettonica vers. 04*, Meletti et al., 2000, shown in Figure 1), used for the PS4 (*Pericolosità Sismica vers. 04*) hazard model (Slejko et al.,

1998). ZS4 was delineated interpreting the seismicity in terms of tectonic regimes and, as a second order criterion, the spatial variation of seismicity; the model was constituted by 80 area sources.

At the beginning of the 21st century, an updated version of the area-source zoning (ZS9; Meletti et al., 2008, shown in Figure 1) was defined for the elaboration of the MPS04 seismic hazard model (Stucchi et al., 2011). ZS9 resulted from modifications, merges and eliminations of the numerous areas delineated in the previous zoning ZS4, as well as from the introduction of new ones. The goal of ZS9 was to build a model consistent with new data collected at the time of its development.

Since most of that knowledge was considered still reliable during the development of ZS9, this latter was built without introducing substantial novelties as regards the general kinematic framework on which ZS4 was based. In some cases, groups of area sources of ZS4 were merged on the basis of the characteristics of the kinematic domain to which each of the area sources was attributed. In the meantime, the geometry of the area sources was modified according to the changed seismotectonic knowledge. Most importantly, in ZS9 area sources were designed strictly enveloping the seismogenic sources that were at that time known and defined in the DISS database (Basili et al., 2008). In ZS4, on the contrary, the areas extended over the known seismogenic sources, including regions where faults were not mapped, according to what was thought to be a cautionary criterion. We should consider that, for PSH assessments using an area source model, the seismicity rates computed using faults and earthquakes located inside the area are equally spaced in a grid of point sources where each point has the same seismicity occurrence properties (i.e. rate of events generated). The arbitrary increase in the surface of some of the area sources of ZS4, therefore, leaded to a reduction of the hazard estimate of PS4 in those areas. ZS9 was then developed by constraining the geometry of the area-sources to the location of seismogenic faults and historical and instrumental earthquakes, avoiding arbitrary extensions of the dimensions of the area-sources. Figure 1 shows the ZS9 model, consisting of 36 area sources, together with ZS4.

The development of ZS16 was driven by the choice to update the area-source model of ZS9 only where new data suggest different interpretations. To summarise, as shown in Figure 1, ZS16 updates ZS9 and constitutes the base for the MA4 ERF. MA4, along with other ERF models, was used as one of the inputs of MPS19 (Meletti et al., 2021; Visini et al., 2021).

## 3. Input data

Area sources for PSHA represent regions with seismicity spatially uniform in terms of earthquake occurrence rates, maximum magnitude, expected rupture mechanism, and so on. In our model, mapped active faults played a major role in defining the boundaries of the area sources, however we integrated geological data with historical and instrumental seismicity, as well as with geophysical data, including geodetic strain field, maximum horizontal stress ($Sh_{max}$) orientation and focal mechanisms, to define the geographical borders of the areas, the prevailing style of faulting, the upper and lower bounds of the characteristic seismogenic depth, and the distribution of hypocentral depths.

To determine the boundaries and the seismic parameters of the area sources we collected and analysed several seismotectonic datasets (Fig. 2), some of which were compiled in the framework of MPS19 (Meletti et al., 2021) to be used as common inputs

for the development of all the ERFs. Among these datasets, we used: an historical earthquake catalogue (*Catalogo Parametrico dei Terremoti Italiani* CPTI15, version 1.5, hereinafter CPTI15; Rovida et al., 2016; 2020); an instrumental earthquake catalogue (Gasperini et al., 2016; Lolli et al., 2020); the version 3.2.1 of the Database of Italian Seismogenic Sources (DISS 3.2.1; Basili et al., 2008; DISS Working Group, 2018); an harmonised GPS velocity model for the Mediterranean area (Devoti et al., 2017); and other geological and geophysical data, available for specific regions and for the whole territory, as described in the following. Data used to draw boundaries of area sources of ZS16, earthquake catalogues and a brief description of the area sources are available on request from the corresponding author.

### 3.1. Earthquake catalogues

CPTI15 v.1.5 lists 4389 earthquakes with moment magnitude Mw $\geq$ 4.0 or macroseismic intensity $\geq$ 5 that occurred in the Italian and neighbouring areas (Fig. 2a) in the period 1000-2014. The catalogue provides epicentral locations and homogeneous Mw estimates derived from both macroseismic and instrumental data. The catalogue takes advantage of the wealth of macroseismic intensity data related to both historical and recent earthquakes collected in the Italian Macroseismic Database DBMI15 (version 1.5; Locati et al., 2016). The parameters of 43% of the earthquakes are calculated from such intensity data with the Boxer algorithm (Gasperini et al., 1999; 2010) with an updated calibration (Rovida et al., 2020). Following Gasperini et al. (2012), instrumental magnitudes are moment tensor solutions complemented with proxy Mw obtained from magnitude estimates in other scales according to Gasperini et al. (2013) and Lolli et al. (2014; 2015; 2018). For the sake of homogeneity, the Mw of modern earthquakes with both macroseismic and instrumental magnitudes is the mean of the two estimates. Although different methods for identifying mainshocks are available in the literature, within the MPS19 project the widely used and tested procedure by Gardner and Knopoff (1974) with the space and time windows defined therein, was selected. The procedure resulted in a catalogue of 3353 mainshocks, corresponding to 76% of the whole CPTI15, that was used in all the ERFs of MPS19.

To define the seismogenic layers and the depth distributions of the earthquakes in the area sources, we also considered an instrumental catalogue with homogeneous Mw determination for the period 1981-2015 that contains about 256000 events without any lower magnitude threshold (Gasperini et al., 2016). An updated and slightly different version of this catalogue was later published by Lolli et al. (2020), with a thorough description of the input data and selection criteria. The catalogue contains the instrumental part of CPTI15, complementing it for magnitude < 4.0.

### 3.2 Focal mechanisms

To collect a representative dataset useful to define the styles of faulting of each area source, we started from the CMT Italian Dataset (Pondrelli et al., 2006; CMT Italian Dataset available at https://doi.org/10.13127/rcmt/italy). It includes all available moment tensors for the Italian peninsula and surrounding areas from 1976 to present with a minimum Mw of 4.0, collected from the Global CMT (Centroid Moment Tensor) Catalogue (Ekström et al., 2012 and references therein) and the Euro-Mediterranean RCMT (Regional Centroid Moment Tensor) Catalogue (Pondrelli and Salimbeni, 2015;

https://doi.org/10.13127/rcmt/euromed). To reach the best homogeneity in terms of spatial distribution, we included in our dataset the moment tensors obtained through seismic data inversion for few Mw ≥ 4.0 earthquakes occurred in the Alpine region, not available in the CMT or RCMT Catalogues, selected from the German Research Centre for Geosciences (GFZ) and ETH Zurich datasets (Saul et al. (2011) and Bernardi et al. (2004), respectively). In addition, to get a dataset with a longer time-coverage, we considered first-motion polarity focal solutions for relevant events that occurred before the digital era, such as the 1968 Belice (Sicily) earthquakes. They have been selected from the EMMA database (Database of Earthquake Mechanisms of the Mediterranean Area; Vannucci and Gasperini, 2004). When multiple focal mechanisms were available for a single event, the choice followed the quality evaluation given in the EMMA database, where a "preferred" solution is defined (see Pondrelli et al., 2020 for details). The final entire dataset, shown in Figure 2b, consists of 995 focal mechanism solutions for events in the magnitude Mw range between 4 to 7.

### 3.3 Active faults

Active faults played an important role in defining the boundaries of the area sources, to this aim we consulted databases referring to different scale of resolution, from national to local scale, to include both the general seismotectonic picture and the details in the draw of boundaries.

The DISS database (Basili et al., 2008; DISS Working Group, 2018) is a fundamental product for interpreting the relationships between faults and earthquakes in Italy. DISS 3.2.1 contains 127 Individual Seismogenic Sources (defined as a simplified and three-dimensional representation of a fault plane. Individual seismogenic sources are assumed to exhibit "characteristic" behaviour with respect to rupture length/width and expected magnitude), 188 Composite Seismogenic Sources (defined as simplified and three-dimensional representations of crustal faults containing an unspecified number of seismogenic sources that cannot be singled out, Fig. 2c), 35 Debated Seismogenic Sources, and 3 subduction zones. All sources are based on geological/geophysical data and cover the whole Italian territory and portions of adjacent countries and seas.

At the national scale, we also considered the "Structural Model of Italy" (CNR, P.F. GEODINAMICA 1990) and the seismotectonic model by Meletti et al. (2000). The latter was used as a guide for identifying homogeneous domains of active tectonics in Italy.

In some regions, we integrated the above datasets with data from local detailed geological-structural investigations to define the boundaries of the area sources, for example: Delacou et al. (2004) and Sue et al. (2007) for northwestern Italy; Collettini and Barchi (2002), Boncio et al. (2004), Papanikolaou and Roberts (2007), Lavecchia et al. (2007a), Faure Walker et al (2010; 2012), Visini (2012), Tesson et al. (2016) and Valentini et al. (2017) for central and southern Italy; Lavecchia et al. (2007b), Catalano et al. (2010), Billi et al. (2010), Visini et al. (2010) and Mastrolembo et al. (2014) for Sicily.

### 3.4 Other geophysical data

As a proxy for evaluating the thickness of the crust and defining zones with similar seismogenic thickness, we used the Moho maps by Solarino and Cassinis (2007) and Di Stefano et al. (2011), and the heat flow maps by Della Vedova et al. (2001).

We also considered the regional strain rate fields for the Mediterranean area derived from GPS data (Devoti et al., 2017) and the maximum horizontal stress Sh$_{max}$ orientation (Mariucci and Montone, 2020) to qualitatively check the homogeneity of the strain rate values and of the Sh$_{max}$ orientations within the area sources.

## 4. The MA4 seismogenic model

MA4 is based on the seismogenic zoning ZS16, which represents the evolution of previous area-source models proposed in the last 30 years as unique area-source input in seismic hazard assessment performed in Italy. The criteria for defining ZS16 are described in Section 4.1, methodologies for calculation of parameters of area source, useful for PSH, are described in Section 4.2 - top and bottom depth of the seismogenic layer, Section 4.3 - hypocentre distribution, and Section 4.4 - style of faulting. MA4 also models seismicity rates for each area source of ZS16 (Section 4.4) and include epistemic uncertainties in the assessment of the completeness intervals of earthquake catalogue (Section 4.4.1), maximum magnitude (Section 4.4.2) and alternative frequency-magnitude distributions (Section 4.4.3).

### 4.1 The ZS16 seismotectonic zoning

Although area sources are widely used for national and international PSHA, there are no standard objective approaches for defining their boundaries. We acknowledge the criteria defined in previous studies (e.g. Giardini, 1999; Meletti et al., 2008; Wiemer et al., 2009; Vilanova et al., 2014; Danciu et al., 2018) to set guidelines for the delineation of area sources in order to describe the correlation between active faults, earthquakes and other geophysical inputs. To update the existing reference national zoning scheme ZS9 we applied the following criteria:

a) start from the area sources of the ZS9 model;

b) be consistent with the general background delineated by the geodynamic model proposed by Meletti et al. (2000), i.e. an area source should belong to a unique tectonic zone (active shallow crustal, volcanic or subduction zone in the specific Italian case);

c) incorporate all recent advances in the understanding of the active tectonics of the territory and in the distribution of seismogenic sources modelled in the DISS 3.2.1 database and other active fault compilations at the national and regional scale (see section 3.3). In particular, for defining area source boundaries that primarily follow the surface projection of mapped active faults: an area source should not interrupt a normal or reverse fault system unless major differences are observed (changes in stress orientation and/or changes in crustal depth); for strike-slip faults, boundaries should be parallel to the strike of the faults and the area source should contain the faults;

d) incorporate information derived from the investigation of the most recent seismic sequences that struck Italy after the compilation of ZS9, namely the L'Aquila 2009, Emilia 2012 and Amatrice-Norcia 2016 sequences;

e) be consistent with the spatial pattern of seismicity depicted by the CPTI15 earthquake catalogue. Area sources whose borders were drawn using mapped active faults (point c) should not cross spatial clusters of earthquakes which are

attributed to the same faults. As earthquakes locations, both macroseismic and instrumental, are affected by uncertainties, a spatial shift between the possible causative faults and the epicentre can occur;

    f)   consider for the definition of the boundaries: the pattern of seismicity, focal mechanisms, geodetic strain field, $Sh_{max}$ and heat flow data;

    g)   account for the variation of the style-of-faulting and tectonic regime with depth, therefore multiple area sources can overlap on the volume domain;

    h)   cover the entire Italian territory, as required by MPS19.

Applying these criteria to the data described in section 3, we defined the seismogenic zoning shown in Figure 2d, consisting of 48 active shallow crustal area sources and 2 area sources corresponding to the Campanian and Mt. Etna volcanic districts (i.e., area sources #31 and #49, respectively). For the deep seismicity related to the Tyrrhenian subduction intraslab, no area sources were defined, because MPS19 adopted a separate ad-hoc ERF for modelling such seismicity. Finally, it is worth noting that 3 area sources (#19, #20 and #25) showed a different kinematics for shallow and deep seismicity, so they have been split and computations for them have been done apart (Electronic Supplement 2, see Pondrelli et al., 2020 for details).

### 4.2 Top and bottom depth of the seismogenic layer and hypocentral distributions

Seismogenic depths for each area source of ZS16 were estimated using the instrumental catalogue by Gasperini et al. (2016). In particular, we assumed the upper and lower limits of the seismogenic layer as corresponding to the 5th and 95th percentiles of the depth distribution of the earthquakes inside each area (e.g., Boncio et al., 2009; Stucchi et al., 2011) and we modelled the depth distributions of hypocentres with the peaks of unimodal and bimodal distributions that best approximate the observed values.

To estimate these values, we first removed the earthquakes with fixed hypocentral depth (i.e. 0, 5 or 10 km), that represent ~10% of the total. We then considered only the earthquakes whose depth is shallower than the Moho, based on the crustal models by Solarino and Cassinis (2007) and Di Stefano et al. (2011). We calculated the (rounded) 5th and 95th percentiles of the hypocentral depth distributions, respectively assumed as the upper and lower limits of the seismogenic layer. As regards area sources #19, #20 and #25, showing a different kinematics for shallow and deep seismicity (Pondrelli et al., 2020), we divided them into "shallow" and "deep" area-sources (i.e. #19s and #19d, #20s and #20d, i.e. #25s and #25d). Based on crustal thickness and rheological properties, we adopted ad hoc values of depth for these area-sources: for shallow area-sources, we assumed depth values of 0 km, 15 km and 10 km, respectively for the top and the bottom of the seismogenic layer and for the hypocentral depth; for deep area-sources, we assumed depth values of 15 km, 30 km and 23 km, respectively for the top and the bottom of the seismogenic layer and for the hypocentral depth.

For depths within the top and the bottom of the seismogenic layer, we computed modal values, standard deviation and log-likelihood of the unimodal and bimodal distributions that best fit the observed values (see Figure 3 for an example). We evaluated and compared the AIC (Akaike Information Criterion) index of the unimodal and bimodal distributions to select the best model for the hypocentral depth distribution of each area. In case of unimodal distribution, we used the modal value as

representative of the hypocentral depth, while for bimodal distributions, we assigned weights to both modal values by using their mixing proportion value in the bimodal distribution. To evaluate the stability of the results with respect to the number and the magnitude of the considered events, we calculated the upper and lower seismogenic depths and the modal values of the distributions for different minimum magnitudes (from Mw 2.5 to 4.5), and compared the resulting depth estimates with the depth of the composite seismogenic sources of DISS 3.2.1 inside the area. In Figure 3, we show an example of the results obtained for area-source #24. We retain that the modal values can be representative of the hypocentral depths, however, future researches could better detail correlations among depth distributions, magnitude and kinematics.

In the Mt. Etna region, we assigned earthquakes with hypocentral depth < 10 km to the volcanic domain (area source #49 in Fig. 2d) and earthquakes with hypocentral depth ≥ 10 km to the underlying active crustal area sources (#44, #45, and #46). For the Campanian volcanic area (#31 in Fig. 2d), we adopted depth values of 0 km, 4 km and 1 km, respectively for the top and the bottom of the seismogenic layer and for the hypocentral depth, mainly based on the parameters of the 2017 Mw 3.9 Ischia earthquake (De Novellis et al., 2018). Electronic Supplement 1 lists the parameters derived for all the area sources for minimum magnitude Mw 2.5, which appears to be the most appropriate threshold value to ensure a significant number of earthquakes for all the areas.

### 4.3 Style of faulting

Pondrelli et al. (2020) defined the criteria to parametrize the styles of faulting of expected earthquake ruptures and to evaluate their representativeness in each area source. Using available seismic moment tensors for relevant events (Mw ≥ 4.5), first-motion focal mechanisms for less recent earthquakes, and also geological data on past activated faults, we collected nearly a thousand of data for seismic events that occurred in the last ∼100 years in the Italian peninsula and surrounding regions, as described in section 2.2. On this dataset we applied in each seismic area-source a procedure that starts with the separation of all available focal mechanisms into the three main tectonic styles, following the rake-based criteria given in Akkar et al. (2014): rake between −135° and −45° defines a normal solution, while between 45° and 135° are reverse solutions, all the rest is classified as strike-slip. We summed all data within each group, i.e. the normal, the reverse and the strike-slip one. Using the results of the summations, in each area-source we identified, when possible, a nodal plane, considering the different percentages of styles of faulting and, where necessary, including a total or a partial (even in terms of tectonic style) random source contribution. Following these steps, we obtained the resulting styles of faulting for each area-source reported in Figure 4 and also in the Electronic Supplement 2 (Pondrelli et al., 2020). As stated above, in a few cases, changes in tectonic style with depth were identified (area source #19s and #19d, #20s and #20d and #25s and #25d).

### 4.4. Annual rates of earthquake occurrences

To estimate the expected seismicity rates of each area source, we adopted a time-independent (i.e. Poisson) model for earthquake occurrence. We assumed that the distribution of the earthquake magnitudes follows the Truncated Gutenberg-Richter (TruncGR, Ordaz, 2004) model that has three parameters: $\Lambda_0$, that is the cumulative number of earthquakes per unit

time equal to or larger than the magnitude threshold or minimum magnitude ($M_{min}$) and smaller than the upper (or maximum) magnitude ($M_u$) and the slope ($\beta$, $\beta = 2/3\ b$). The TruncGR distribution is the Pareto distribution with the probability density function truncated at both ends. Its cumulative density function related to moment magnitude is:

$$\Lambda_{(M)} = \Lambda_0 \frac{e^{-\beta Mmin} - e^{-\beta M}}{e^{-\beta Mmin} - e^{-\beta Mu}} \tag{1}$$

$\beta$ and $\Lambda_0$ were derived from the declustered CPTI15 catalogue (see section 3.1) by adopting the completeness time intervals described in the following (section 4.4.1) and applying a maximum-likelihood fit based on Weichert (1980). $M_u$, the upper magnitude, is described in the section 4.4.2. The minimum magnitude for the application of Weichert (1980), $M_{min}$, is estimated following the maximum curvature approach.

### 4.4.1. Completeness time intervals

Two independent sets of completeness time intervals for the CPTI15 catalogue were defined according to i) the historical approach of Stucchi et al. (2004; 2011), and ii) the statistical method proposed by Albarello et al. (2001).

The historical approach determines the complete intervals analysing the local history of a set of sample localities. Based on this knowledge, the years from which it is unlikely that earthquakes effects of a given intensity are not recorded in the local historical sources were determined. The catalogue can be considered as complete for earthquakes of the same epicentral intensity ($I_0$) located at or near the analysed locality. Stucchi et al. (2004; 2011) assessed the starting year of completeness for intensity $\geq 6$ at 18 sample localities and then extrapolated them to the area sources they belong to and to others with similar history and seismotectonic features. As this approach is independent from the catalogue, we used the same completeness intervals of Stucchi et al. (2011) for $I_0 \geq 6$, adapting the 5 macro-regions defined therein to ZS16. In these macro-regions we also evaluated the completeness intervals for $I_0$ 4-5, 5 and 5-6 with the same criteria, and we assessed all those of a newly-introduced offshore region ("Sea" in Fig. 5a). The completeness intervals determined in this way were then applied to Mw bins defined according to the new empirical conversion relation between epicentral intensity and magnitude of CPTI15 (Rovida et al., 2020). The estimated Mw bins are of 0.23 Mw units, which corresponds to the difference between the Mw values obtained from the discrete $I_0$ values, including also uncertain intensity assessments as "half degrees", e.g. 6-7 equal to 6.5. This choice avoids the uneven concentrations in the same bin of Mw values derived from the conversion of different discrete epicentral intensity values of historical earthquakes.

The statistical completeness intervals were assessed using the procedure of Albarello et al. (2001) for the same macro-regions defined for the historical approach. The method is very sensitive to the number of earthquakes considered in each area and magnitude bin, and to ensure the stability of the results we selected magnitude bins of 0.46 Mw units, which means grouping in the same class integer and intermediate intensity values (e.g. intensity 6-7 together with 6). The results were then applied to the same bins of 0.23 Mw units defined for evaluating the historical completeness. The completeness starting years for each

$I_0$/Mw bin and macro-region defined according to the two approaches are shown in the Electronic Supplement 4. Figure 6 shows a comparison, in each macro-region, of the completeness periods and magnitudes assessed with the historical and statistical approaches. The complete catalogues resulting from the historical and the statistical assessments are made of 2496 and 2603 earthquakes, i.e. 63% and 66% of CPTI15 (excluding earthquakes of the Etna volcanic area and subduction events of the Calabrian Arc), respectively.

The two methods, although with differences due to their assumptions, provide comparable results within each macro-area. The difference in the numbers of events in the complete periods arises from the intervals assessed with the statistical approach for low magnitude bins, which contain more events than highest ones, longer than those obtained with the historical approach. On the contrary, the historical approach determines longer complete periods for the highest Mw bins, which are less rich in events (Fig. 6).

In conclusion, taking also into account the declustering procedure mentioned above (Section 3.1), the catalogues used for calculating annual rates of earthquake occurrences within the MA4 model contain 1800 and 1888 events, obtained respectively with the historical and the statistical completeness assessment.

### 4.4.2. Maximum magnitude

For the definition of the maximum magnitude we used the estimates provided for MPS19, described in Visini et al. (2021), which were based on the estimate of the maximum observed earthquake in the earthquake record from CPTI15. The Italian area was divided into 18 tectonic domains (Fig. 5b) and the earthquakes listed in CPTI15 were assigned to them, according to their location. Based on the average error of magnitude estimates for earthquakes occurring before and after 1980, a minimum value of uncertainty for the Mw evaluation of 0.3 and 0.2 was introduced for the historical and instrumental portion of the catalogue, respectively. The maximum observed magnitude inside a tectonic domain is the largest magnitude observed, including the uncertainty ("$Mw_{obs}$+uncertainty"). In low seismicity regions (for example, those located along the Tyrrhenian coast and in the north of Italy), albeit relative abundance of low magnitude earthquakes, the earthquake record spans over a few hundred years only, and this time interval may be much smaller than the recurrence time of moderate earthquakes. As a result, the maximum observed magnitude in these regions relies on a poor historical record not useful to constrain the upper magnitude. In these situations, Visini et al. (2021) adopted analogies to similar tectonic features (Wheeler, 2009) or cautionary approaches based on expert judgement of minimum values of maximum Mw. Then, following Woessner et al. (2015), a minimum value of maximim magnitude, $Mw_{max}$, was assigned to each tectonic domain ($Mw_{tect}$), namely 6.5 for all active crustal areas, 6.0 for the Tyrrhenian tectonic domain, and 5.6 for the volcanic area of Etna. Two values of maximum magnitude were then assigned to each tectonic domain: i) $Mw_{max1}$ (Figure 5c), that is the largest value between $Mw_{tect}$ and $Mw_{obs}$+uncertainty, and ii) $Mw_{max2}$, that results by uniformly incrementing $Mw_{max1}$ by a cautionary value of 0.3 to account for epistemic uncertainties, except for the Etna volcanic domain where $Mw_{max2}$ is equal to $Mw_{max1}$. Furthermore, where active faults are known, Visini et al. (2021) compared the magnitude estimates of the composite seismogenic source (CSS) by the DISS 3.2.1 with the $Mw_{max1}$ and $Mw_{max2}$ from earthquake catalogue in each area-source. As already pointed out in Visini et

al. (2021), caution should be posed in comparing the earthquake catalogue based and the DISS based magnitudes. The $Mw_{max}$

of the CSS (namely $Mw_{css}$) is computed based on empirical regressions, or from the literature, or by associating the largest historical earthquake of the area to the CSS. In the latter case, however, there is not necessarily a correspondence between the magnitude associated with a CSS in DISS 3.2.1 and the one provided by CPTI15, as a result of the different and independent development of the two databases. For two tectonic domains $Mw_{css}$ exceeds the $Mw_{max2}$. For tectonic domain 2 (Fig. 5b), $Mw_{css}$ was evaluated using the Mw of the 1690 earthquake as derived from Guidoboni et al. (2007; 2019), whereas the Mw of the

same earthquake in CPTI15, including uncertainty, is ~6.5 and it is located in the tectonic domain 3. In this case, Visini et al. (2021) decided to anchor $Mw_{max1}$ to the $Mw_{tect}$, considering that Mw 6.5 corresponds to the magnitude estimated in the most recent CPTI15 catalogue. In the tectonic domain 12, the $Mw_{css}$ is the Mw 6.8 2003 earthquake, based on the modelling from Meghraoui et al. (2004) of the coseismic uplift. Because $Mw_{max2}$ agrees with the original estimate of the magnitude of the 2003 earthquake, Visini et al. (2021) kept the $Mw_{max2}$. (see Visini et al., 2021 for details).

$Mw_{max1}$ and $Mw_{max2}$ are used as upper bounds ($M_u$) of the earthquake recurrence of various magnitudes specific to each seismogenic source.

### 4.4.3 Seismic rates determination

To calculate annual rates of earthquake occurrences for the active shallow crustal areas, we first excluded from the CPTI15 the events belonging to the Southern Tyrrhenian subduction (i.e. those located in that area with hypocentral depth larger than

40 km), and we imposed a minimum of 10 earthquakes and at least 2 non-empty classes of magnitude (0.1 bin size) in each area source to derive stable $\beta$ values; otherwise, we assumed $\beta$ equal to 2.3 (b =1).

We used 5 different approaches to calculate the seismicity rates for the area sources, which are based on 2 different assumptions.

The first assumption is that $\beta$ varies across the areas, then a first approach (named, approach i) consists in the classical

estimation of the parameters $\Lambda_0$ and $\beta$ (see Eq. 1) directly for the area sources, from the earthquakes belonging to each source with magnitudes between $M_{min}$ and the maximum magnitude ($Mw_{max01}$ or $Mw_{max02}$, "statistical" or "historical" completeness according to the branch of the logic tree).

The second assumption is that β is stable over groups of area-sources with similar seismotectonic features. The rationale behind this assumption is to use a sample of earthquakes as rich as possible to estimate a robust β-value, while maintaining a

relationship between the slope of TruncGR and the seismotectonic characteristics of a region (for example area-sources characterised by extension). Indeed, for area sources characterised by a "low" rate of seismic activity, the $\beta$ (or the b)-value can be biased and results as an artefact of the low number of data available (e.g., Marzocchi et al., 2020). To maintain β variations from area to area, however, we used the approach i (described above), in which the $\beta$-value is actually estimated for each area-source. Under this second assumption, we first calculated $\beta$ and $\Lambda_0$ for groups of area sources, hereinafter defined as

"macroarea", using earthquakes belonging to each macroarea with magnitudes between $M_{min}$ and the maximum magnitude (as for the approach i, $Mw_{max0}1$ or $Mw_{max02}$, "statistical" or "historical" completeness are taken according to the branch of the logic

tree). To this purpose, we used the tectonic regions shown in Figure 5b. As the next step, we assessed the recurrence parameters in each area-source within a macroarea. Keeping with our objective to only change $\Lambda_0$ according to the level of seismic activity of an area, we defined 4 different approaches for partitioning $\Lambda_0$ to the sources belonging to the same macroarea. In particular,

defining $\Lambda_{0as}$ and $\Lambda_{0ma}$ respectively as the values of $\Lambda_0$ of the area source and of the macroarea, the four approaches are:

ii) In each macroarea, we compared the observed seismicity rates of occurrence of the macroarea with the ones of the area sources. We compared seismicity rates of magnitudes ranging between $M_{min}$ (estimated for the macroarea) and $M_u$. For each area source, and for each magnitude, we calculated the ratio between the observed seismicity rates of occurrences of the macroarea and of the area source. For each area source, we calculated the average of these ratios. Finally, we used the average

ratios of all the area sources of a macroarea to proportionally scale $\Lambda_{0ma}$ to $\Lambda_{0as}$;

iii) to build the TruncGR of each area source in a macroarea, we used $\beta$ and $M_u$ of the macroarea, whereas $\Lambda_{0as}$ is firstly assumed to be the observed rate of exceedances for magnitudes equal or larger than $M_{min}$ proper of the area source. Then, we scaled the expected rates of the TruncGR of each area source in a macroarea, to ensure they sum up to the TruncGR of the macroarea they belong to.;

iv) For each area source in a macroarea, the TruncGR uses $\beta$ and $M_u$ of the macroarea. The $\Lambda_{0as}$ is computed at the $M_{min}$ of the macroarea and it is the value that maximises the log-likelihood of the observed and expected (forecasted) number of earthquakes in each area source. In particular, for each magnitude of an area source, we computed the expected number of earthquakes by multiplying the seismicity rate of occurrence (not in the cumulative form) with the corresponding interval of completeness. Then, for each area, we sum the log-likelihood obtained for each magnitude and find for the $\Lambda_{0as}$ that allows to

obtain the best log-likelihood)

v) For each area source in a macroarea, the TruncGR was computed using $\beta$ and $M_u$ of the macroarea. As regard $\Lambda_{0as}$, for each area source, we used the observed rates of exceedance for magnitudes equal or larger than the $M_{min}$ proper of each area source, and computed $\Lambda_{0as}$ at the $M_{min}$ as the value that minimises the root-mean-square of the observed vs expected seismicity rates (given by the TruncGR).

For the areas #19, #20 and #25, we successively partitioned the 5 $\Lambda_{0as}$ between the "shallow" and "deep" sources, according to their relative percentage of number of earthquakes with Mw $\geq$ 2.5 (values are in the Electronic Supplement 3).

Figure 7 shows an example of the frequency-magnitude distributions calculated directly for the area sources #24 and #36 and for, respectively, the macroareas #6 and #11. Whereas in areas of relative "high" rate of seismicity (as the extensional areas in the Central Apennines , area-source #24), evaluations of $\beta$-values are approximately stable with both assumptions (between

approach i and approaches ii-to-v), in area characterised by a "low" rate, the differences in the estimates of $\beta$-values can lead to variations in the expected rates modelled by the TruncGR, as for the area-source #36 in the strike-slip regions in the Apulian foreland. As we cannot know a-priori what is the best solution between $\beta$-values variable at the level of the single areas or at the level of large tectonic domains, we embedded the 5 approaches in a logic tree approach (described in the next section and shown in Fig. 8). In the Electronic Supplement 3, we listed the parameters of the TruncGR distributions associated with each

area source.

## 5. Seismic hazard estimation using MA4

Seismic hazard was calculated over the whole Italian territory (including sites located within 5 kilometres outside the borders) adopting the MA4 seismogenic model. To this purpose, 52 area sources were used, considering the area-sources #19, #20 and #25 in the form "shallow" and "deep" and discarding the area-source #49 (Etna). In fact, MPS19 defined additional ad-hoc

ERFs for 3 specific regions: a) the Etna volcanic area, which replaces the ERF for area-source #49; b) the subduction shallow interface seismicity and deep intra-slab seismicity of the Calabrian Arc (spanning from the Ionian Sea to the southern Tyrrhenian Sea across the Calabria region); and c) the seismogenic sources located outside the area of the CPTI15 catalogue (see Meletti et al., 2021; Visini et al., 2021).

Alternative choices and interpretations about the key elements were embedded in a logic-tree structure (Kulkarni et al., 1984;

Coppersmith and Youngs, 1986; Senior Seismic Hazard Analysis Committee, 1997), that is the conventional tool to capture the epistemic uncertainty associated with the input elements of a PSHA model. The adopted logic tree, shown in Figure 8, consists of a first branching level accounting for the 2 alternative evaluations of the catalogue completeness time intervals described in section 4.4.1, i.e., one based on historical information (Stucchi et al., 2004; 2011) and one on the statistical approach by Albarello et al. (2001). Then, a second branching level considers the 2 alternative sets of maximum magnitude

$Mu$ (eq. 1) values (i.e. $Mw_{max1}$, $Mw_{max2}$), described in section 4.4.2, and a third level accounts for the 5 approaches adopted for calculating the frequency-magnitude distributions of each area source (see section 4.4.3). Since we have no specific reason to prefer or to differently weight the 20 resulting ERF branches, the same weight (i.e., 1/20) was assigned to each of them. Following Woessner et al. (2015) and Danciu et al. (2018), we considered the uncertainties on hypocentral depths and focal mechanisms as aleatory and we used distributions and weights described in the previous sections to model them. Two further

branching levels account for alternative choices of GMMs, as selected and applied in MPS19, that are 3 GMMs for active shallow crustal regions (Bindi et al., 2011; Bindi et al., 2014; Cauzzi et al., 2015, with associated weights equal to 0.45, 0.32 and 0.23, respectively), and 1 model for volcanic areas (Lanzano and Luzi, 2020). The selected GMMs provide estimates of ground shaking in terms of the geometric mean of the horizontal components. In particular, the 3 models adopted for shallow crustal regions derive from a pre-selection of 16 candidate GMMs performed over nearly one thousand models published in

the literature. Then the performance of the pre-selected GMMs was evaluated through the comparison with accelerometric records available for Italy and Europe using different scoring methods; a final rank of the 16 candidate GMMs was done and the 3 best performing models were selected; finally, the weights of the 3 GMMs were assigned combining the results of the scoring and the weights obtained through an experts' elicitation (see Lanzano et al., 2020 for details). Besides showing the best relative hindcasting skill, the 3 selected GMMs use different metrics for the distance (i.e., Joyner–Boore distance, $R_{JB}$, for

Bindi et al., 2011, hypocentral distance, $R_{hyp}$, for Bindi et al., 2014, and distance from the rupture plane, $R_{rup}$, for Cauzzi et al., 2015) and are calibrated on different datasets (i.e., Italian, European, and global, respectively). As a result, we obtained a final logic tree made of 60 branches (Fig. 7).

Seismic hazard was calculated for rock-site conditions (i.e., EC8 site category A or $Vs_{30} \geq 800m/s$) using the OpenQuake engine platform (Pagani et al., 2014). We recall here that, for the computation of seismic hazard, OpenQuake discretizes every area source into a regular grid of points, each representing the longitude and latitude of the centre of a rupture. Every rupture has a rectangular shape and is centred on the hypocentral distribution parameterized in ZS16. Ruptures are created using the magnitude-area scaling relationship of Wells and Coppersmith (1994) and by conserving an aspect ratio of 1. The aspect ratio is adjusted if the rupture plane would grow beyond the upper and lower seismogenic depths, in order to keep the area.

Figure 9 illustrates the spatial distribution of mean Peak Ground Acceleration (PGA) values obtained by applying the weighting scheme in Figure 8, for 10% and 2% probabilities of exceedance in 50 years, also referred to as 475-yr and 2475-yr return periods, respectively. The map for PGA at 10% probability of exceedance in 50 years (Fig. 9a) shows the highest hazard estimates (PGA $\geq$ 0.25 g) in 3 areas, i.e., in the northeast of Italy, along the central Apennines, and the southern Apennines. Hazard levels ranging between 0.1 g and 0.25 g characterise the majority of coastal areas of the Adriatic and Tyrrhenian seas, the southern part of the Po Plain, Sicily, and large parts of southern Italy. The northwest of Italy, Sardinia and part of Apulia show hazard values generally lower than 0.075 g.

The map for PGA at 2% probability of exceedance in 50 years (Fig. 9b) shows that PGA $\geq$ 0.5 g characterises almost entirely the Apennines, part of the southern Po Plain and the northeast of Italy. The southeast of Sicily shows hazard levels ranging between 0.3 g and 0.5 g, whereas PGA ranging between 0.2 g and 0.3 g characterises the rest of Sicily and the majority of the Adriatic and Tyrrhenian coastal areas. Hazard levels lower than 0.1 g are obtained in the northwest of Italy, in the southern part of Apulia and in Sardinia.

Figures 10 and 11 respectively show hazard curves for PGA and Uniform Hazard Spectra (UHS) for spectral periods from 0.1 s to 4 s for the cities of Milano, L'Aquila, and Siracusa (see locations in Fig. 9), chosen for exemplificative purposes. Figure 10 illustrates the variability of the expected ground motions in PGA, showing the mean hazard level (black line), the hazard curves resulting from each of the 60 realisations (grey lines) and the uncertainties expressed through the 16th and 84th percentiles (red lines). Figures 11 show the mean and the 16th and 84th percentiles of UHS for 10% and 2% probability of exceedance in 50 years.

Figure 12 shows the spatial distribution of the coefficient of variation (CoV) of PGA values for 10% and 2% probabilities of exceedance in 50 years. Recalling that GMMs were weighted and the 60 branches do not have the same weights, we calculated the CoV as the weighted standard deviation divided by the weighted mean. CoV < 0.2 cover a large part of the Italian territory, indicating a low uncertainty in the expected acceleration. CoV > 0.2 characterise the areas with the lowest PGA and the south-eastern sector of Sicily. The latter corresponds to the area-sources where the $\beta$ values show the largest differences between the macroarea and the single area-source approaches.

The CoV values in Figure 12 contain uncertainty related to both the ERF and the GMMs. We then investigated the relative contribution of both components of epistemic uncertainty at 3 selected localities. For each site, hazard curves for PGA were plotted by distinguishing the 3 groups composed of 20 realisations that use the same GMM for active shallow crustal regions. In Figure 13, the 3 groups are shown by light-red lines for the realisations that adopt Bindi et al. (2011), light-green lines and

light-blue lines for those using Bindi et al. (2014) and Cauzzi et al. (2015), respectively. Figure 13 also shows the mean Probability of Exceedance (POE) in 50 years for the 3 groups. It can be seen that most of the seismic hazard curves from the different GMMs are overlapping, and the uncertainty due to the GMMs is of a similar order of magnitude as the uncertainty related to the ERF. For Milano and L'Aquila, the mean curves of the 3 groups overlap in a wide range of POE, especially for POE < 10 % in 50 years. For Siracusa the Bindi et al. (2011) group intersects the Cauzzi et al. (2015) one at POE of ~ 10% - 11% in 50 years; for PGA < 0.02 g, the realisations returning the highest POE are those using the latter GMM and the realisations returning the lowest POE are those using Bindi et al. (2011). This trend is inverted for PGA > 0.05 g or POE < ~ 12% in 50 years.

To focus on the relative contributions of ERF and GMM uncertainties, in Figure 14 we illustrate the POE for a series of levels of PGA, by distinguishing the branches that used a particular GMM. In particular, in Figure 14, the POE for each level of PGA shown, is normalised to the maximum, in order to have all values ranging from values > 0 and <= 1, for ease of visualisation. Normalisation does not affect next considerations, as we are not interested in describing absolute POEs, but relative contributions. Figure 14 allows, in a qualitative way, to describe the relative importance of the uncertainty due to ERF modelling in respect to the GMM contribution. We plotted branches that use the same GMM with the same colour: red for those branches that used Bindi et al. (2011); green for Bindi et al. (2014), and blue for Cauzzi et al. (2015). If the uncertainty due to GMMs impacted more than the ERF modelling, we would expect a cluster of values grouped by the same colour (GMM). This is not the case shown in Figure 13, on the contrary, the symbols of different GMMs are distributed to cover, approximately, the same ranges. The overlaps of values occur at all the PGA shown and for the three localities, we interpret this as an indication of the importance of ERF modelling that can contribute in an equal, even if not larger, manner than the GMMs.

## 6. Discussion

The MA4 seismogenic model in Italy represents a challenge for the construction of an area-sources based model that takes into account the variety of seismotectonic environments, which include spatial and depth variations of the main style of faulting. The MA4 model is based on a seismotectonic zoning (ZS16), defined according to a list of criteria specifically define to this purpose. However, we cannot exclude that the proposed zoning still contains some a priori bias or, simply, some area-sources could not reflect the actual tectonics. Objective criteria to delineate area sources with a quality ranking of the basic data would be an additional step (e.g. Wiemer et al., 2009; Vilanova et al., 2014).

As an example, the debate on "large" vs "small" areas concerns subjective choices. "Small" areas are designed to capture changes in seismicity at the local scale (e.g. < 20 km) but, in our opinion, these changes are better highlighted with fault-based models or a smoothed seismicity approach. Our idea of zoning aims at individuating the areas that have a homogeneous behaviour from a seismotectonic point of view, focusing on the similarities rather than on the differences. The experience in seismogenic zoning in Italy since the early 1990s (e.g. the ZS4 model by Meletti et al., 2000) suggests that small area-sources,

such as to represent single faults, can be appreciated by structural geologists, but make defining their earthquake recurrence parameters very difficult because of the lack of available data that can even produce apparent differences in seismicity distribution at the local scale. Nevertheless, the size of an area source delineating a pattern of high seismicity should be sufficiently small, otherwise the high rate of seismicity is distributed over a larger area-source and thus is dangerously reduced due to the effect known as "spatial smearing" (National Research Council, 1988). Therefore, seismic hazard results are different if the same quantity of seismicity is assigned to sources of different size: the smaller the source area, the higher the resulting hazard estimates, and vice versa.

The ZS16 seismotectonic zoning was recently also adopted in the new European Seismic Hazard Model (ESHM20) as the reference area source model for Italy (Danciu et al., 2021).

To determine the activity rates of area-sources from earthquake data we initially used the concept of macroarea to evaluate some parameters, such as the maximum magnitude and the β-value of the truncated GR relation, which were then assigned to all the area-sources belonging to the macroarea. This approach serves to prevent biased spatial variations of b-value due to poor datasets. The uncertainty in magnitude and epicentral location was assumed to be minimal with respect to the other sources of uncertainty in the estimation of seismic hazard, such as maximum magnitude, completeness time intervals or the choice of the GMMs. However, uncertainty related to the magnitude-location-depth estimation of the earthquakes propagates throughout the evaluation of the completeness periods and thus impacts the estimation of seismic rates.

MA4 explored a range of sources of epistemic uncertainty and is constituted by 20 alternative ERFs, that consider different options in terms of completeness time intervals, maximum magnitude and earthquake rates estimation. We observed that the uncertainty related to this set of ERFs is comparable to or even higher than the uncertainty related to alternative GMMs. Although this observation is not generalizable to other ERFs, it contributes to the discussion on the relative importance of ERFs and GMMs in the overall uncertainty affecting seismic hazard estimates. The GMMs adopted in MPS19 were selected according to statistical criteria and elicitation procedures (Lanzano et al., 2020). We did not test the single ERFs (or branches) of MA4 with statistical procedures, but the mean seismic rates and the associated uncertainty were positively checked against observations (i.e. number of earthquakes occurred in the last centuries) before entering MPS19 (see Meletti et al., 2021 for details). In our opinion, however, trimming ERFs according to their performance against observations contributes to reducing the epistemic uncertainty, but could result in a selection of models that is only guided by the earthquake occurrence realisation observed in the past.

## 7. Conclusions

In this study, we presented a new seismogenic model for Italy, called MA4, that is part of a community-based effort that led to the development of a new seismic hazard model for Italy (MPS19; Meletti et al., 2021). MPS19, in fact, involved more than 150 Italian researchers at various stages of the project and many of them were involved in the building of their earthquake rupture forecast (ERF). The finally adopted 11 groups of ERFs, were composed of a number of alternatives that explore the

epistemic uncertainty of seismicity modelling (see Visini et al., 2021). In this framework, MA4 represents one of the 5 area-source seismicity models included in the set of ERFs.

The MA4 model is based on an update (ZS16) of the previous seismogenic zoning ZS9 (Meletti et al., 2008) adopted by the current Italian reference seismic hazard model MPS04 (Stucchi et al., 2011). The new seismogenic zoning consists of 48 active
shallow crustal area sources and 2 area sources corresponding to the Campanian and Mt. Etna volcanic districts (this last not used for computation here shown).

We used 5 different approaches to calculate the expected seismicity rates for the area sources, based on 2 different assumptions. The first assumption is that the truncated Gutenberg-Richter relation parameter $\beta$ varies across the areas and the second assumption is that $\beta$ is stable over groups of area-sources with similar seismotectonic features. For this reason, we determined
the seismicity rates by first calculating the truncated Gutenberg-Richter parameters for groups of area-sources (macroareas) considered tectonically homogeneous, and successively partitioning in different ways the values to the area-sources forming each macroarea.

In conclusion, MA4 consists of 20 seismicity models that consider epistemic uncertainty in the estimations of the completeness periods of the earthquake catalogue, of maximum magnitude values and of seismicity rates. The results show that the
uncertainties explored by the 20 different seismicity models is at least of the same order of magnitude of the uncertainty due to the different GMMs. We encourage future studies to account for such uncertainty propagation.

**Author contributions.** FV wrote the draft of the paper, prepared the codes for the analysis and ran the hazard calculations. All the Authors contributed to design the boundaries of ZS16, to parametrize the MA4 model, to revise seismic hazard results
and to write the final version of the manuscript.

**Data Availability.** Data used to draw boundaries of area sources of ZS16, shapefile of ZS16, earthquake catalogues and a brief description of the area sources are available from the corresponding author on reasonable request.

**Acknowledgments.** We acknowledge the availability of Michele Simionato to help us with OpenQuake and Francesco Martinelli for the development of the server and cluster for OpenQuake.

This study has benefited from funding provided by the Italian Presidenza del Consiglio dei Ministri – Dipartimento della Protezione Civile (DPC). This paper does not represent DPC official opinion and policies.

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

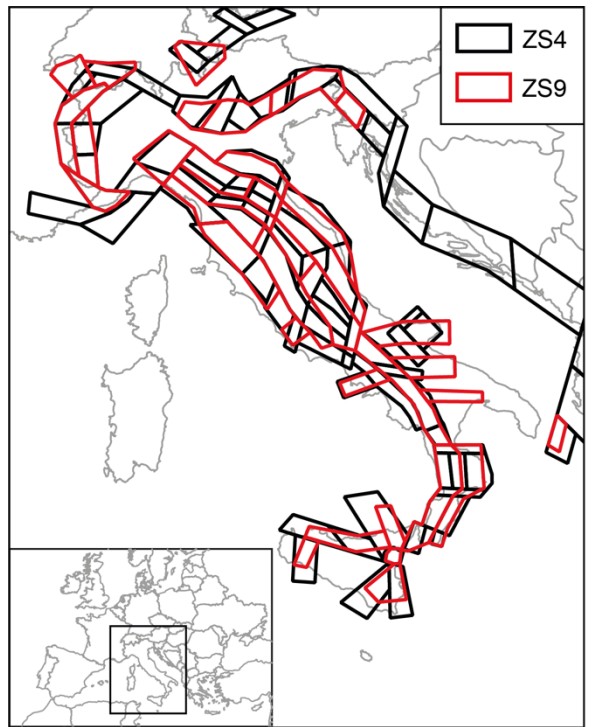


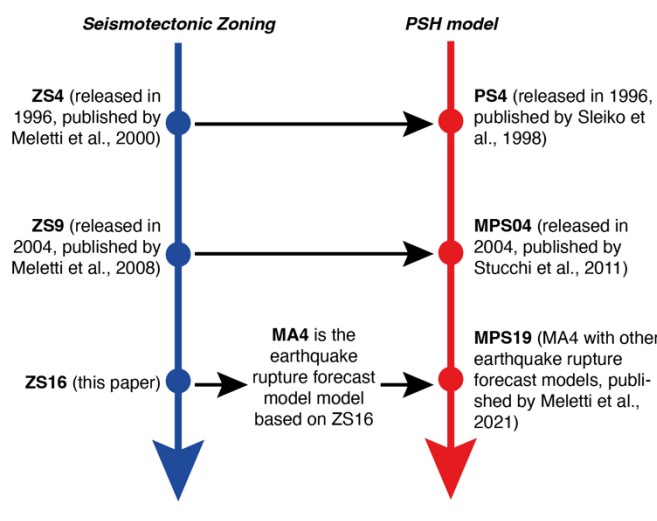

**Figure 1. To the left, comparison of ZS4 by Meletti et al. (2000) and ZS9 by Meletti et al. (2008). ZS9 is the seismogenic zoning of the current Italian seismic hazard model (MPS04) by Stucchi et al. (2011). The sketch on the right shows the relationships between the seismogenic zoning and the seismic hazard models developed in Italy. Note that only for the last ZS16, we distinguished the**
**seismogenic zoning from the earthquake rupture forecast model.**

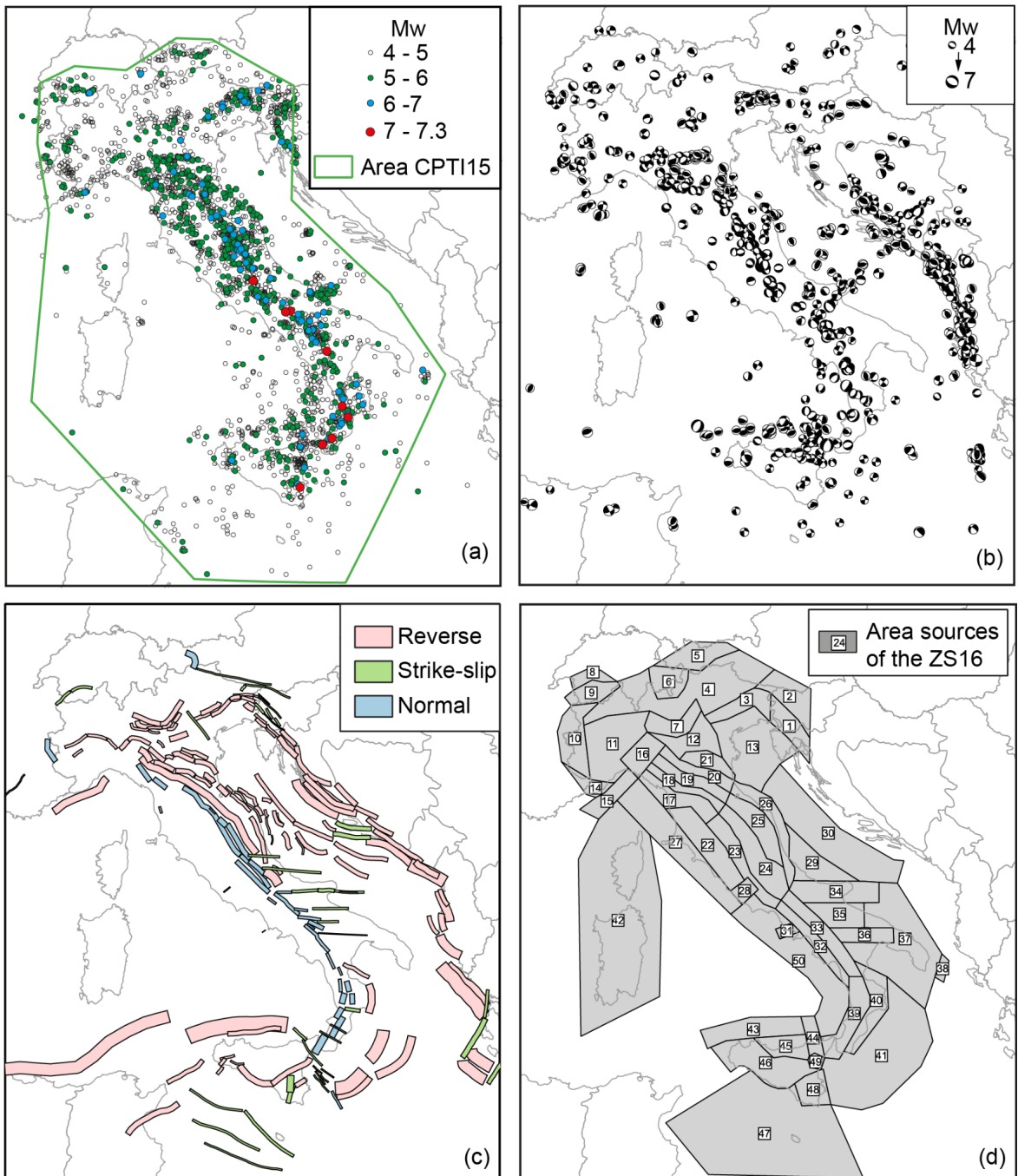

**Figure 2. Main datasets (a, b, c) used to build the ZS16 (d) seismogenic zoning. a) Earthquake epicentres from the CPTI15 catalogue (the green polygon represents the area covered by the CPTI15, as described by Rovida et al., 2016; 2020); b) focal mechanisms of earthquakes with Mw ≥ 4 (Pondrelli et al., 2020); c) Composite Seismogenic Sources from the DISS3.2.1 database (Basili et al., 2008; DISS Working Group, 2018); ; d) Seismogenic zoning ZS16 proposed in this study.**

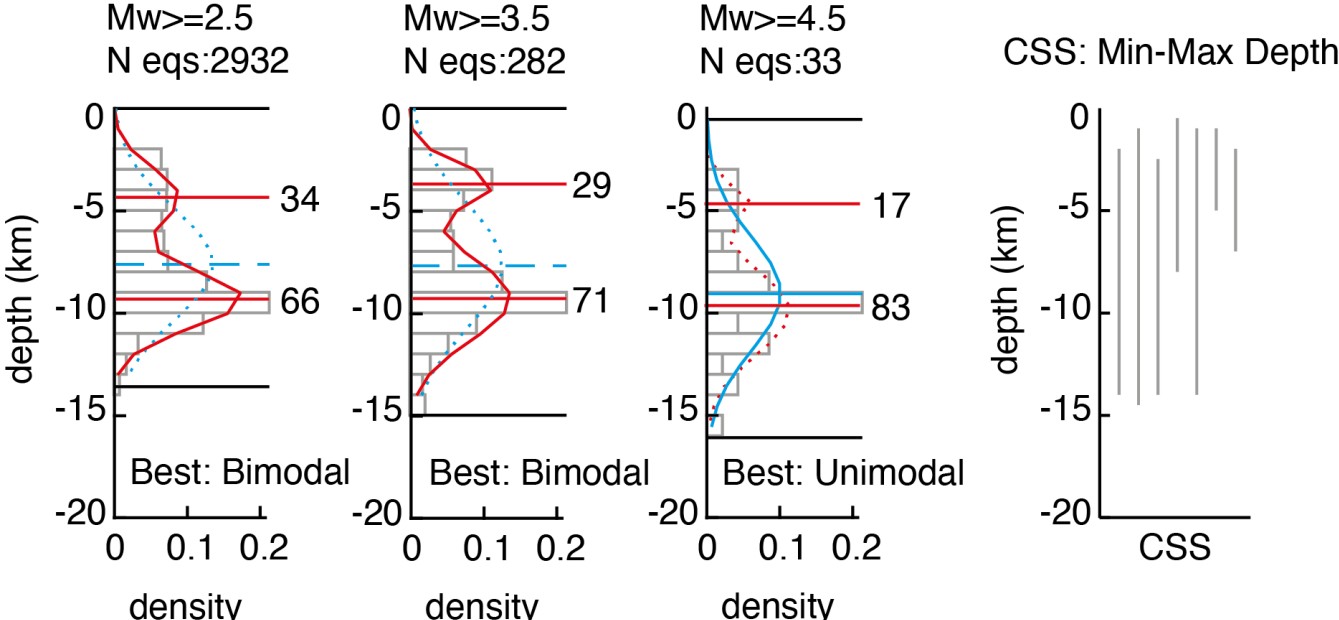

**Figure 3. Hypocentral depth distributions (grey bars) for different threshold magnitudes (reported on top of each panel along with the number of considered earthquakes) for area-source #24. Black lines correspond to the 5th and 95th percentiles, assumed as upper/lower seismogenic depths (round value); blue curve and line represent the unimodal distribution and its modal value; red curves and lines represent the bimodal distribution and its two modal values. Solid lines indicate the best model between uni- and bi-modal distributions, dashed lines the other model. The right panel shows the depth ranges of the composite seismogenic sources**
**(CSS) of DISS 3.2.1 inside the area. The depth of the Moho, from Solarini and Cassini (2007) and Di Stefano et al. (2011) is approximately 30 -35 km.**

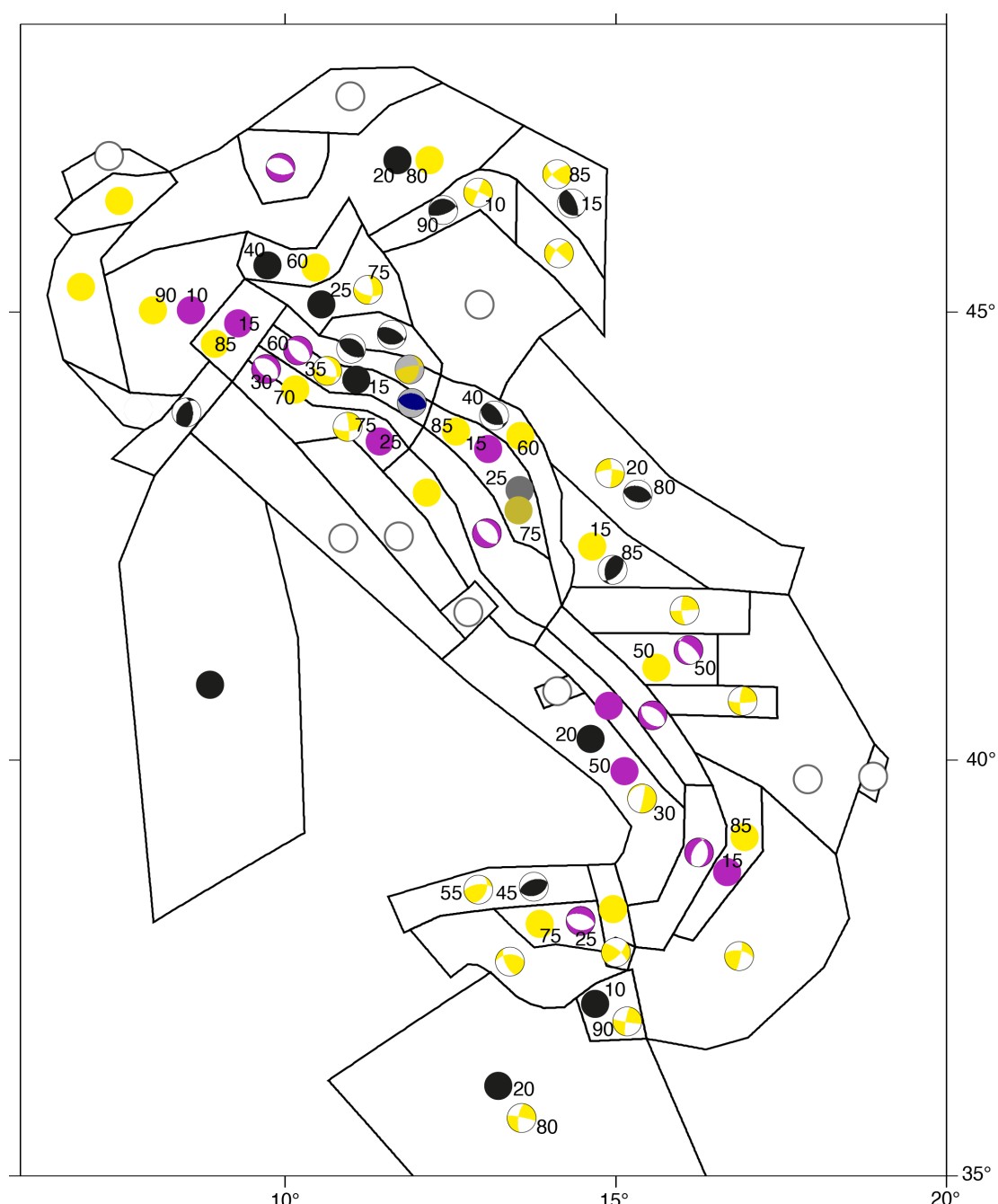

**Figure 4. Expected style of faulting for each area source (modified from Pondrelli et al., 2020). Plain circles represent random seismic sources. White circles represent 100% random while black, purple, and yellow circles correspond to reverse, normal, and strike-slip random sources, respectively. Cumulative focal mechanisms colours follow the same criteria. Focal mechanisms with a grey background or plain circles with darker colours are the sources for deeper layers. Black numbers are the percentages of contribution to the final sources when their sum is the expected style of faulting.**

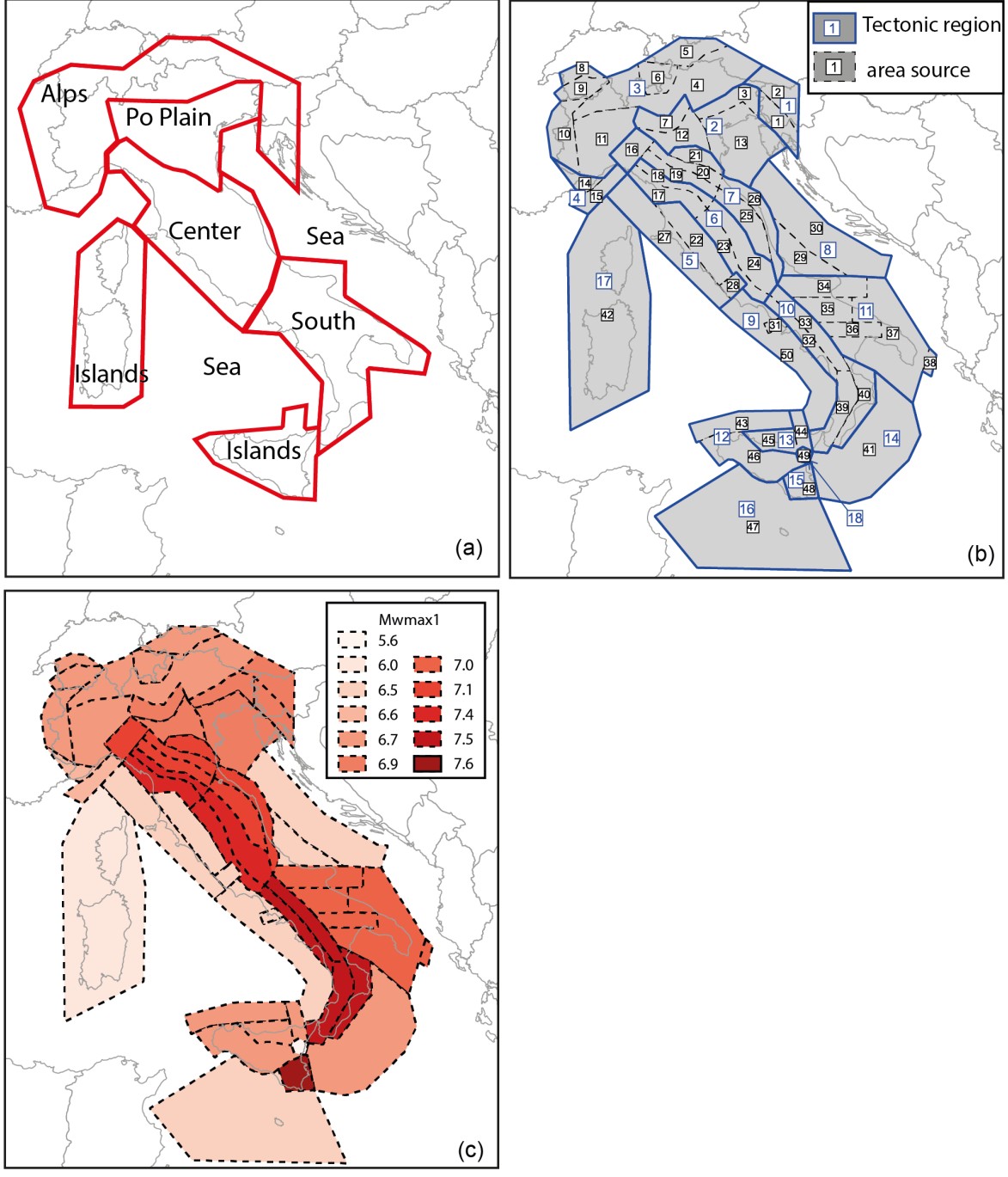

**Figure 5. a)** Macro-regions adopted for evaluating the completeness time intervals for the earthquake catalogue; **b)** Tectonic domains (blue polygons) used to calculate β overlapped to the ZS16 area sources (dashed black polygons); **c)** Map of the values of Mwmax1, $Mw_{max2} = Mw_{max1} + 0.3$.


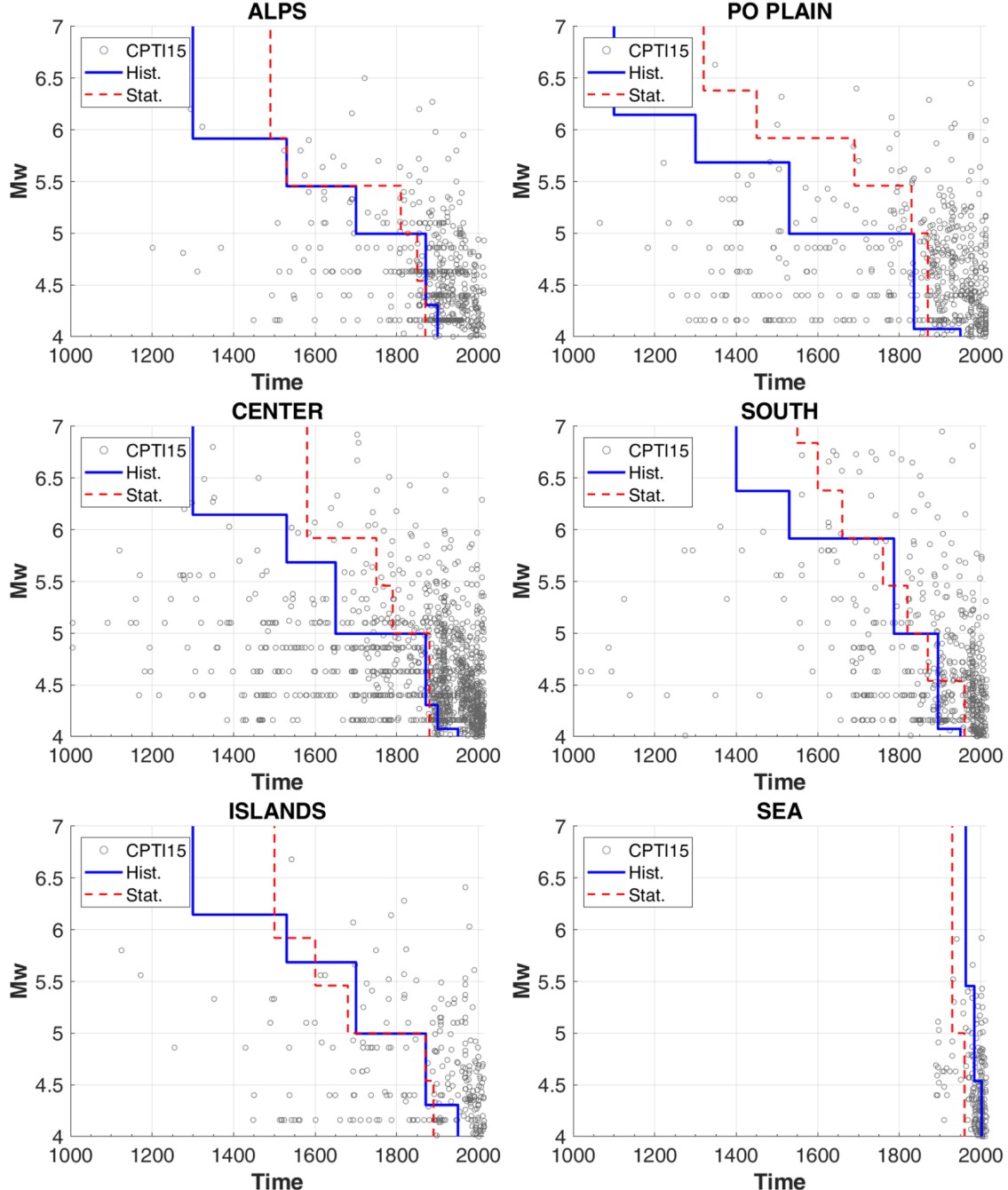


**Figure 6. Plot of time versus completeness magnitude, defined according to both the historical and statistical approach, in the six macro-areas shown in Figure 5a. Grey circles represent the earthquakes in CPTI15. Hist = Historical approach, Stat = Statistical approach.**

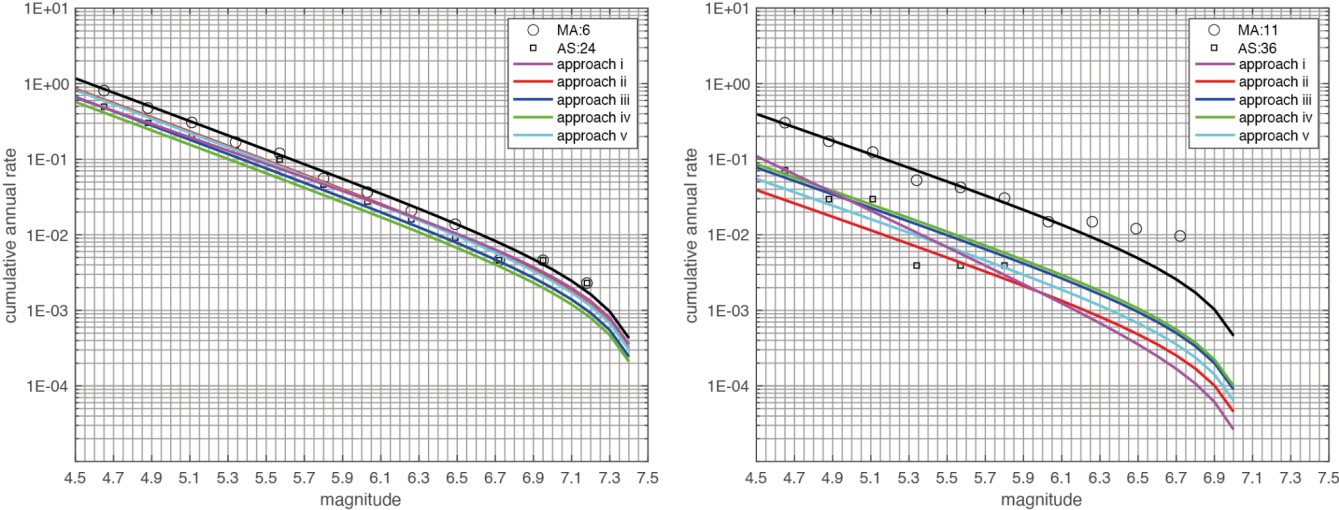

**Figure 7. Example of the frequency-magnitude distributions calculated for the area sources (AS) #24 and #36. The frequency-magnitude distributions of the macroareas (MA) they belong to are also shown. The area source and the macroarea are shown in Figure 5b.**

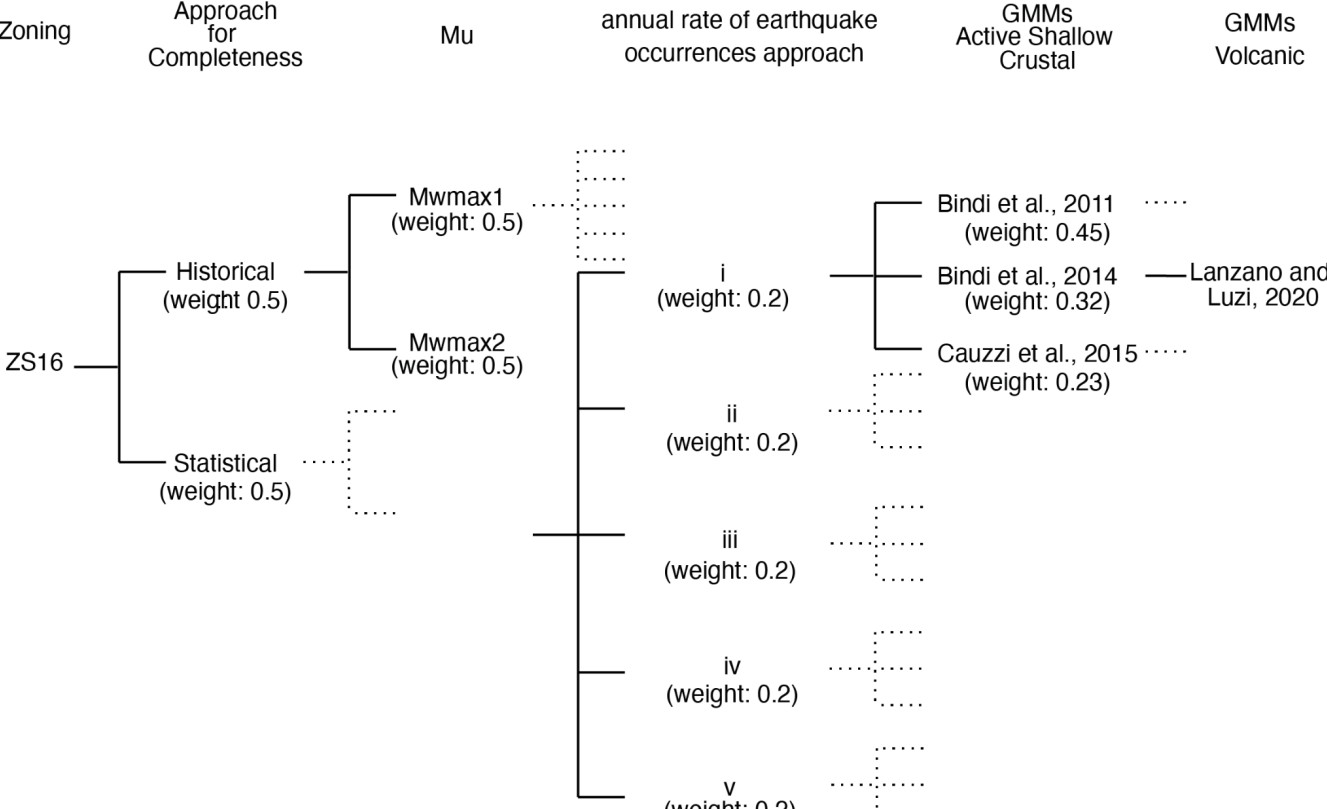

**Figure 8. The logic-tree scheme adopted in this study. "ZS16" is the seismogenic zoning. The completeness time intervals for the CPTI15 catalogue were defined according to the historical approach of Stucchi et al. (2004; 2011) and the statistical method of Albarello et al. (2001). Mu "Mw$_{max1}$" and Mu "Mw$_{max2}$" are the two sets of values adopted for the maximum magnitude, described in Visini et al. (2021). Letters "i"-to- "v" identify the 5 approaches used to calculate the annual seismic rates (see section 4.3). The last two nodes concern the GMMs used for active shallow crustal regions and volcanic areas.**


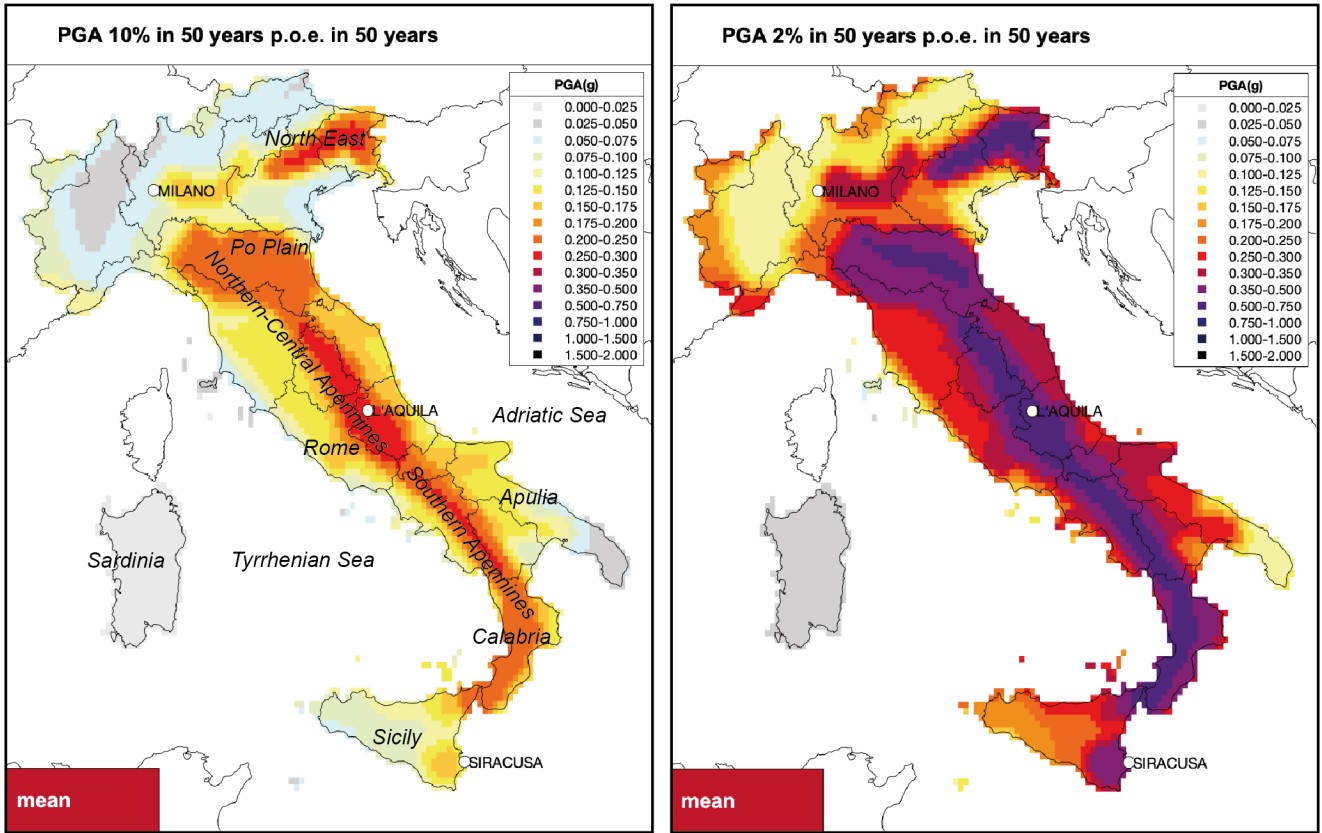

**Figure 9. Maps of mean values of PGA at 10% (left) and 2% (right) probability of exceedance in 50 years. Locations of the 3 cities selected for detailed analyses are shown.**

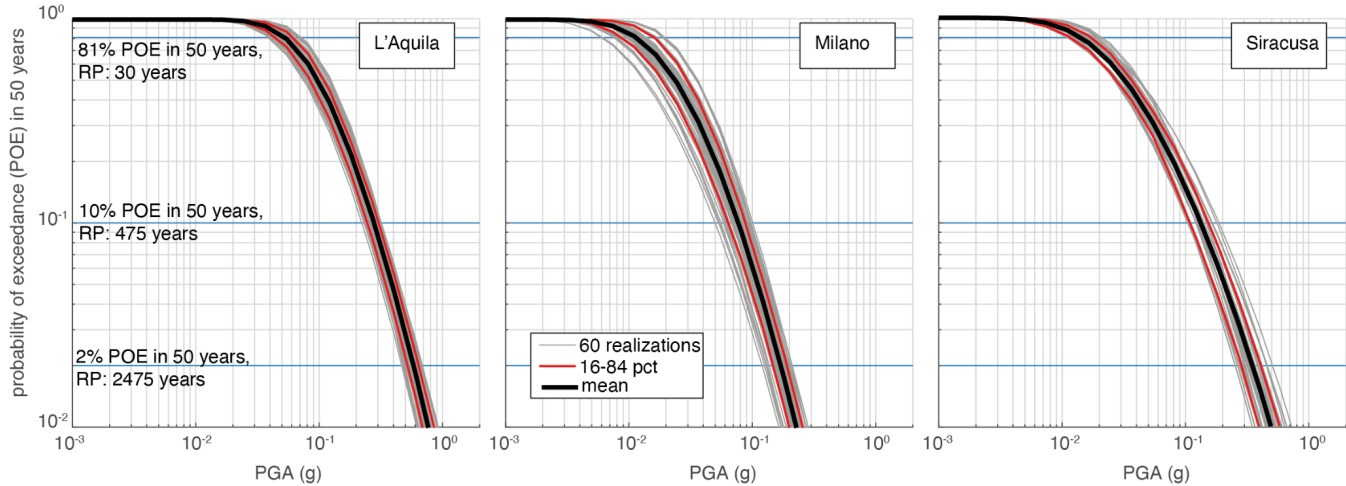

**Figure 10. Hazard curves for PGA for the 3 selected cities. The curves represent: the mean hazard level (black line), the hazard resulting from each of the 60 branches (realisations, grey lines) and the uncertainties expressed through the 16th and 84th percentiles (red lines). The legend in the centre panel refers to all panels.**


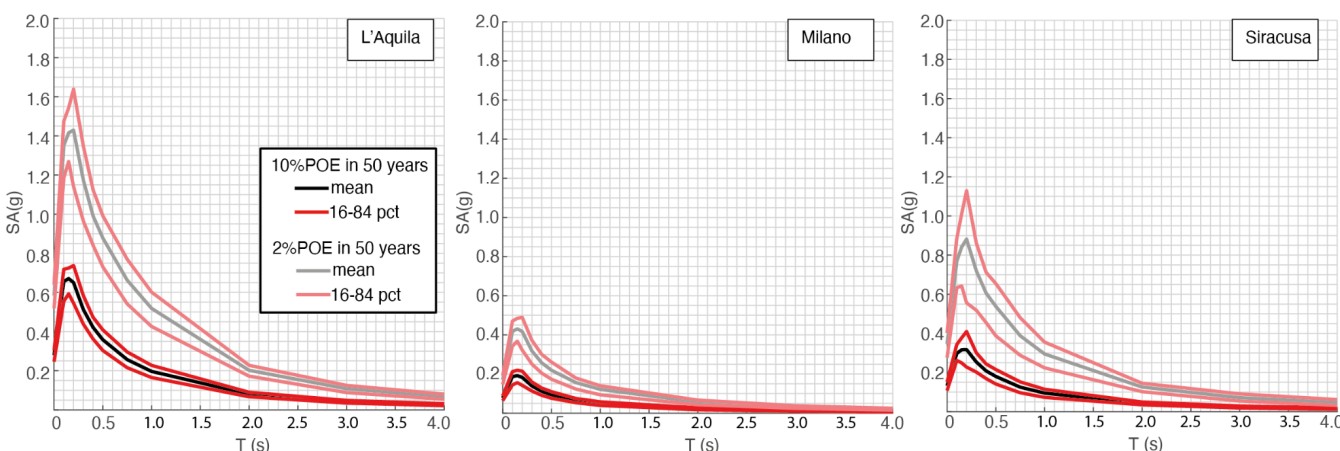

**Figure 11. UHS for 10% (lower spectra, bright colours) and 2% (upper spectra, pale colours) probability of exceedance in 50 years for the 3 selected cities. The mean spectra and the 16th and 84th percentiles are reported. The legend in the first panel refers to all panels.**

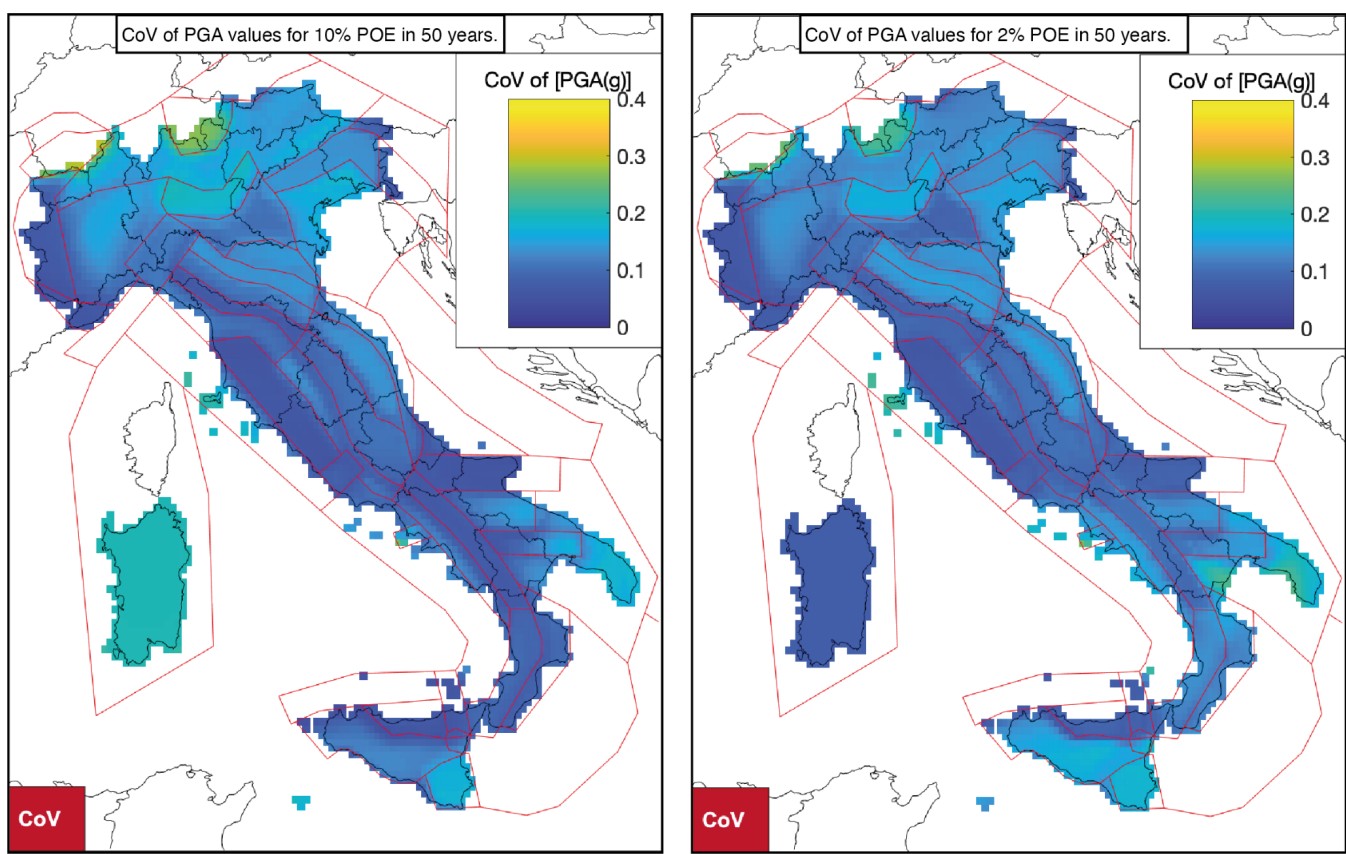

**Figure 12. Spatial distribution of the coefficient of variation (CoV) of PGA values for 10% (left) and 2% (right) probabilities of exceedance in 50 years.**


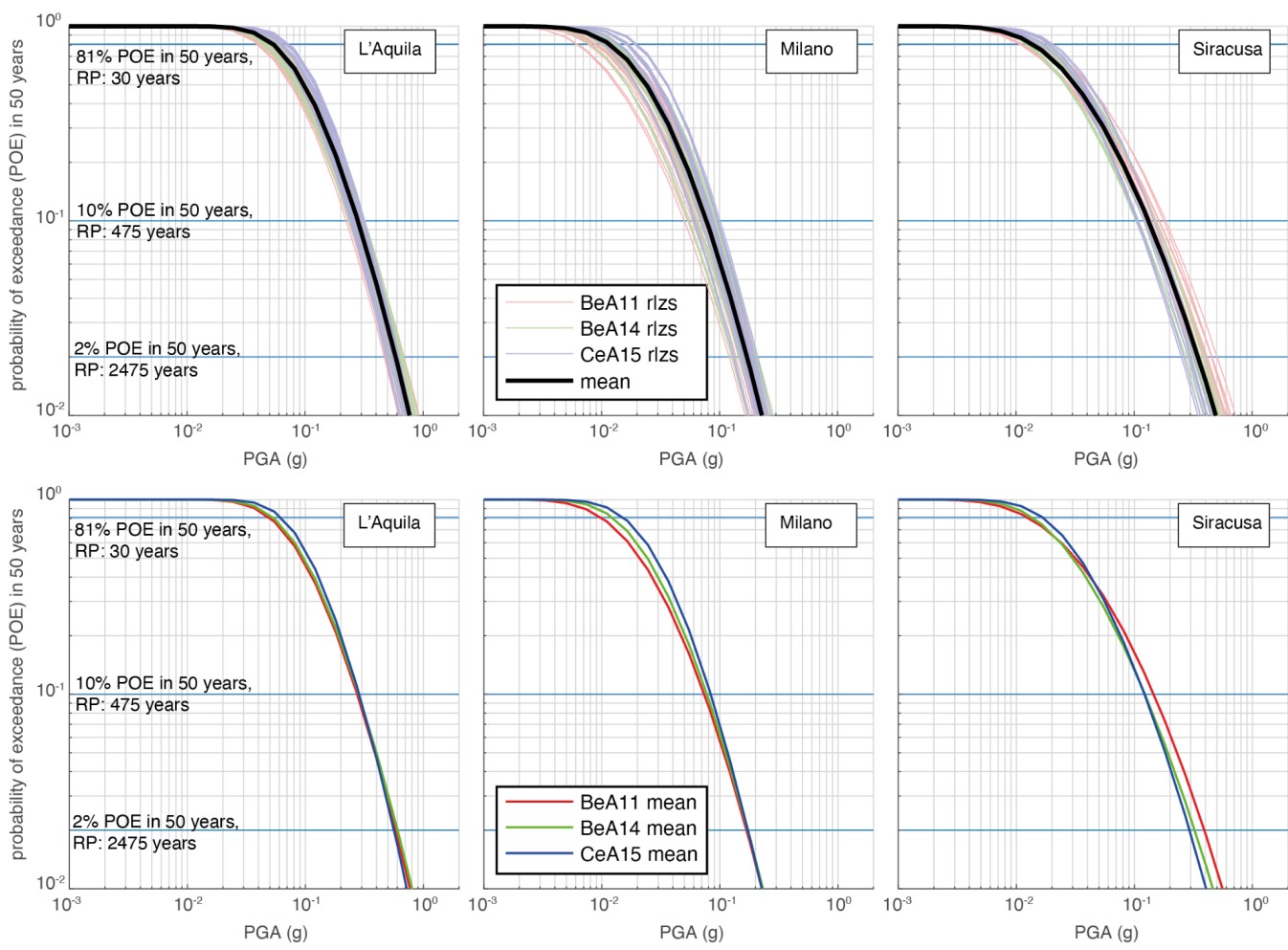

**Figure 13. Hazard curves for PGA for the 3 selected cities. In the first row, the three panels show the mean hazard curves (black lines) and the hazard curve of each of the 60 branches (realisations, coloured per GMM with pale colours). The legend in the centre panel refers to all panels. In the second row, the three panels show the mean hazard curves obtained as a weighted average of the realisations per each GMM (coloured per GMM with bright colours). The legend in the centre panel refers to all panels. Key for all panels: BeA11 = Bindi et al., 2011; BeA14 = Bindi et al., 2014; CeA15 = Cauzzi et al., 2015.**


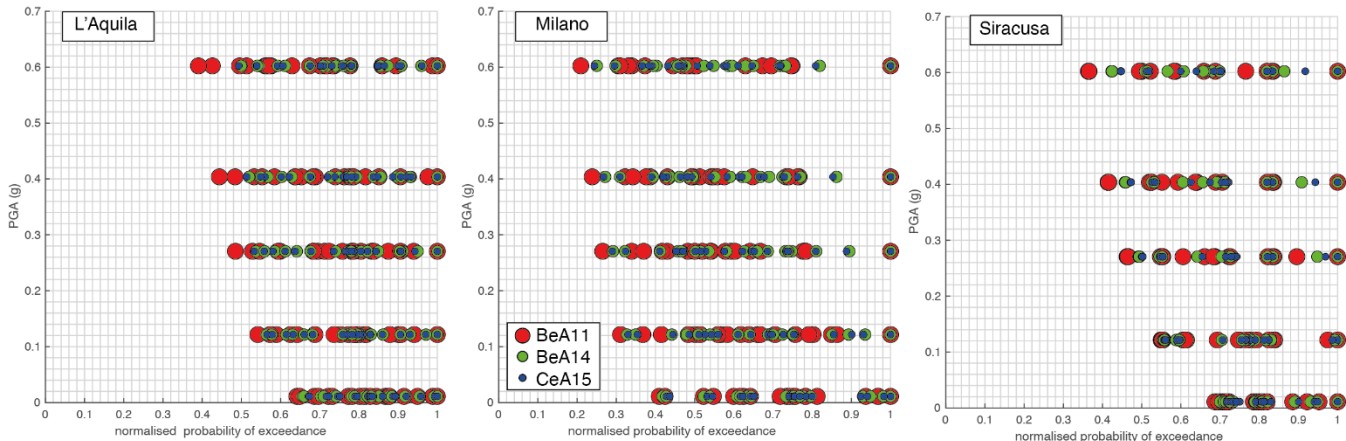

**Figure 14. Relative contributions of epistemic uncertainty in ERF and GMMs hazard estimates. Branches that use the same GMM are shown with 3 colours, red, green and blue. The x-axis shows the POEs normalised to their maximum values for each PGA level. The legend in the centre panel refers to all panels. Key for all panels: BeA11 = Bindi et al., 2011; BeA14 = Bindi et al., 2014; CeA15 = Cauzzi et al., 2015.**
