# Peer review of "An updated area-source seismogenic model (MA4) for seismic hazard of Italy"

_Natural Hazards and Earth System Sciences, 2022_

## Referee Comment (RC3)

[referee-annotated manuscript omitted]

---

## Author Response (AR1)

Dear Referee 1,

Thank you for your useful suggestions and comments. We appreciated the detailed review and we modified the text according to your comments.

In the following we answer point by point.

*REF: Organization of the text: The introduction should be shortened and some of the material and explanations that are currently there should be transferred to a proper section: all the explanations on how ZS9 was established, using ZS4 as a starting point.*

ANSW: We separated the state of the art and explanations on how ZS9 was established adding a separate paragraph. According to REV3 We included a scheme in Figure 1 to explain the link among ZS4, ZS9, ZS16 and MPS04 and MPS19.

*REF: Page 1 in the last decades => decade.*

ANSW: We corrected it.

*REF: Page 2, ZS9 is called ZS4, and Z16 is called ZS9, there is a mix of names which need to be corrected (mix with MPS4 and MPS9?)*

ANSW: We checked for typos. The new paragraph (Section 2) should help to avoid misunderstanding.

*REF: Page 2, "in ZS9 the choice of drawing area sources large enough to include all the seismicity above a certain magnitude threshold, a criterion used in [...] ", this sentence is unclear*

ANSW: We modified this sentence to better explain this point in the manuscript.

*REF: "the increase in the surface of area sources incorrectly reduced the hazard estimate in the central parts of the area" : this sentence is not that clear, why would the largest density of events be in the central parts ?*

ANSW: The drawing of zones, especially for ZS4 and ZS9, sometimes followed the pattern of seismicity, then zones may bound "clusters" of earthquakes. In case of non-homogeneous spatial distribution of seismicity, larger zones produce a lower hazard with respect to zones designed around clustered seismicity. In practice, keeping fixed the number of earthquakes in a zone, the larger the polygon around earthquakes, the lower the density, and this returns a spreader level of hazard.

*REF: Page 3, Section, introduction, It would be nice to see maps that show how you have used the geophysical data to define the areas. This part is quickly treated without much detailed explanations or maps. It is a pity. It is important to understand how area sources are delineated, the process that leads to the polygons.*

ANSW: This is a good point, also raised by the other two reviewers. We decided that all the data (georeferenced maps of the geophysical data and shapefiles) we used to draw the zones can be available by request. We agree that transparency is an important topic, but we would avoid copyrights issues. We also added a brief description for group of areas sources.

*REF: Page 3, section 2.1*

*"Earthquakes [..] that occurred in the Italian and neighboring areas"*

*"The parameters of the 43% of" => the parameters of 43% of*

ANSW: We corrected these sentences.

*REF: Section 3.1 The SZ16 seismotectonic zoning 'be consistent with the CPTI15 earthquake catalogue': what do you mean?*

ANSW: We explained that we refer to the spatial pattern of seismicity. In particular, zone borders drawn using faults should not separate groups of earthquakes attributed to the same faults.

*REF: Section 3.2. This section is difficult to read and should be re-organized for the sake of clarity, e.g. by splitting the section in different paragraphs, beginning the description with the most common sources and then describing volcanic sources.*

ANSW: We changed it as suggested.

*REF: Why using only 1 or 2 depth values, instead of a pdf that would best represent the depth distribution per source zone?*

ANSW: hypocentres were affected by errors in the order of a few kilometres, then the precise shape of the pdf may be uncertain. We retained that the modal values were the most stable representation. In future works, however, we would examine more in detail this part, to analyse correlations among depths, magnitude and kinematics. We added this sentence in the text.

*REF: Prevalent : what do you mean with this term? Is it the most appropriate?*

ANSW: We removed "prevalent" and used the term "representative" where necessary.

*REF: "basing on" => based on*

ANSW: We changed it as suggested

*REF: Section3.4*

*A reference must be indicated for the Gutenberg-Richter model (for the exact equation used).*

*The equation would be more readable in equation format.*

ANSW: We added the reference and changed it as suggested

*REF: Section 3.4.1*

*Do you use exactly the same time windows as Stucchi et al. 2011 ? This is not clear if you apply the method, or use the same windows. I assume you use those from Stucchi et al. 2011, otherwise more information should be provided on how they have been assessed.*

ANSW: We modified the Section in order to better explain that we used the same time windows as in Stucchi et al. (2011) for epicentral intensity I0 >=6 and we defined new ones for I0 < 6. These time windows were then applied to the new macorareas defined according to ZS16 and to the Mw intervals derived from the updated I0 to Mw conversion of CPTI15.

*REF: "Being the approach" ?*

ANSW: We changed it, as also suggested by REV3.

REF: "to avoid the oversampling of some Mw intervals that contain values derived from the *conversion of more than one discrete epicentral intensity value" => this sentence is not clear*

*REF: This is unclear why in the historical method, a 0.23 bin is used; then in the statistical method, 0.46 is used. Strange that the historical method can handle a smaller bin.*

*REF: More would need to be said so that the reader can appreciate the estimation of periods of completeness step.*

ANSW: We detailed and better explain these points in the Section "Completeness time intervals". Following also the suggestion of REV3, we added a plot of time versus completeness magnitude in order to compare the completeness intervals resulting from the two methods. We also added a description of the final catalogue(s) used in the model, i.e. declustered and complete according to the two approaches.

*REF: Section 3.4.2*

*"for the definition of the maximum magnitude, we used the estimates provided by MPS19, described in Visini et al. 2021": can something be said about these estimates ? how they have been evaluated ?*

*Woessner et al. 2015 does not provide the rational behind selecting 6.5 as a minimum for active crustal areas. It would be important to have one explanation for the choice of this value as threshold.*

*"the two values of Mwmax were also checked with the estimates of the maximum Mw of the composite seismogenic sources of DISS ..." : how checked ? What happened if one fault inside the source provided an Mmax larger than the largest magnitude observed including uncertainty ?*

ANSW: We added details on how the maximum magnitude values were estimated in MPS19 (and adopted in our manuscript) and statements on the rationale behind selecting thresholds in active crustal areas. We also discussed the comparison between catalogue-derived maximum magnitude and the one of the composite seismogenic sources of DISS.

*REF: Section 3.4.3 is not that clear. Approach I should belong to one paragraph. Approaches ii to v should be grouped in one paragraph.*

ANSW: We modified the text according to the suggestion.

*REF: ii) observed rates : for which magnitude ?; iii) what is the "threshold magnitude Mt" of an area source?; iv) observed and forecasted number of earthquakes above a given magnitude ?; v) minimize the root-mean-square of observed rates => within which magnitude interval ?*

ANSW: We modified the text and more explanations are now given.

*REF: Section 4, "3 ERFs developed adhoc" : please, what do you mean ?*

ANSW: In MPS19 3 ERFs were developed for i) the Etna volcanic area, ii) the sources external to Italy, and iii) the subduction zone. These three "zones" were then excluded by the 11 seismicity models adopted for MPS19. In our manuscript, we rephrased these sentences to be more clear.

*REF: "the annual rates of earthquake occurrences are given as a non-cumulative magnitude-frequency distribution" => if this detail is provided, then you could provide the name of the distribution in openquake format, otherwise this detail is not necessary*

ANSW: We deleted this sentence, we agree that this detail was not necessary.

*REF: "the cov is the weighted std divided by the weighted mean": why weighted ?*

ANSW: because GMM were weighted, the 60 branches have not the same weights. This implied we cannot use the CoV as simply the standard deviation divided by the mean. However, to be more clear, we rephrased this sentence. We understand that, as it was, it could create misunderstanding on the classical formulation of the CoV.

*REF: Section 5, Discussions and Conclusions*

*Some sentences in the third paragraph are not clear, e.g. :*

*"because of the lack of available data that can even produce apparent differences in seismicity distribution at the local scale"*

*"seismic hazard results are different if the same quantity of seismicity is assigned to sources of different size"*

ANSW: We modified the paragraphs to be more clear. In the first case, we were referring to the fact that few earthquakes in a zone can result in a biased b-value. In the second case, we were referring to the dimension of the zones in respect to the seismicity that is actually contained in. Keeping fixed the a- and b- values, which are determined by the earthquakes inside the zone, the dimensions of a zone

(i.e. the area) impact the density and therefore the levels of seismic hazard. In a point close to the centre of a zone, in fact, the seismic hazard decreases with the increase of the area.

*REF: I fully agree with the last sentence of section 5. We don't know if the future will reproduce observations in the past. We know that the catalogs available are still too short to be representative of what may occur. We should not discard models on the basis that they do not reproduce the past.*

ANSW: Thanks for this point.

*REF: Figure 1. The new zoning should be put on top of the old one. Figure 2. Legend for zoning ZS16 should be put in the graphic that displays the zoning. Figure 3. There is no need to show the y-axis below 20 km depth.*

ANSW: We modified them according to the comments.

*REF: It is not clear where the 5% bar is ? how can it be at zero depth if it is the percentile 5% of the distribution?*

ANSW: We rounded to the nearest integer the value, we missed to specify this, however, we modified the text.

*REF: Figure 4. Caption is very difficult to follow. Full circle: do you mean plain circles?*

ANSW: yes, we mean plain coloured circles. We modified the caption.

*REF: Figure 6. To appraise the impact on the rates of a given method, it would be extremely important to superimpose models for a given source (group by source the 5 alternative results). Also, to understand the variability of the results due to the 5 different methods, it would be very important to show a source with poor data (few events).*

ANSW: We agree and we modified this figure.

*REF: There are too many log scales on these graphics (from 1e-8 up to 100). It produces a visual bias on the alignment of observed rates. There is no need to keep the axis below 1e-5 nor above 10, nor above magnitude 7.5. The annual rates are cumulative or non-cumulative?*

ANSW: We agree and we modified the scales. The annual rates are cumulative.

*REF: Figure 9. If indicating probabilities over 50 years (horizontal lines), it would be clearer to plot probability over 50 years versus acceleration.*

ANSW: We agree and we modified it.

*REF: Figure 10. Final uncertainties are rather small (considering the 16 to 84 percentiles), with respect to other PSHA studies.*

ANSW: This is due to the uncertainty explored in the logic tree. However, we will remark that MA4 is only one branch of the MPS19, and that epistemic uncertainty on the ERF modelling was considered in MPS19.

*REF: To read the graphics, keeping the y-axis ticks on all graphics would help.*

ANSW: We modified the graphics.

*REF: Figure 12. The figure is difficult to read, and it is then difficult to follow the corresponding text page 11. One solution would be to get rid of the individual realization, in order to see the three curves corresponding to the three GMMs. The observation made by the authors that "the uncertainty due to the GMMs is of similar order of magnitude as the uncertainty related to the ERF" is not obvious on this figure.*

ANSW: we modified the figure according to this suggestion in order to render more clear the comparison of the uncertainty related to ERF and GMM

*REF: Figure 13. The cov is calculated from the distribution for a given acceleration level, and this is sound. It would be more straightforward to plot the acceleration versus the cov, rather than a mean probability which has a loose meaning. In any case, mean APO should appear in the y-axis label.*

ANSW: We revised the figure to be easier to be read.

*REF: "Although the scatter in the results for the different sites and PGA levels": verb is missing?*

*Caption : Againt =>against*

ANSW: we adjusted these phrases.

Dear Referee 2,

Thank you for your useful suggestions and comments.

In the following we answer point by point to your comments here.

*REF: - Seismic rates: five "models" are defined and given the same weight in the logic tree. I believe there is not much difference in them from the practical point of view, but I agree that seismic activity is a very important parameter controlling the final hazard values, so it is justified to consider that wide suit of alternatives. I wonder however if the authors could give an opinion in which one actually they believe more. In my opinion I would go for approach one (i).*

ANSW: We appreciated the Reviewer getting this important point. For many zones there is not much difference from the practical point of view, however the seismic activity is a very important parameter controlling the final hazard values. We understand that the Reviewer can prefer the approach 1, as this can catch the finest spatial variation of the GR parameters. From our point of view, the two "most representative" options, from a qualitative point of view, are the approaches 1 and 2.

*REF: - Note that in line 265 you cite "Mt" as magnitude threshold, when elsewhere is cited as $M_0$. Not sure if this is a typo or you actually mean it.*

ANSW: We checked and corrected the use of Mo and Mt through the text (now is Mmin)

*REF: - Style of faulting. I would be very interesting to know which criteria is considered to classify the different styles of faulting. I assume this is done based on the rake, but which rake values have you used to classify the ruptures? Aki criteria? Additionally, for the later calculations with OpenQuake I believe you have to state a fault-plane (strike/dip) for each of the different types (reverse, normal and s-s), which values do you use or are these random?*

ANSW: According to Pondrelli et al. (2020), to which we refer for the nodal planes calculations, to define the three main tectonic styles we adopted the rake-based criteria given in Akkar et al. (2014), which attributes each focal mechanism to either reverse, normal, or strike-slip. In particular, normal solutions have a rake between −135 °and −45°, reverse solutions between 45° and 135°, and other rake values are classified as strike-slip.

*REF: - Hypocentral distribution: This analysis is very nicely performed. The paper states (line 291) that this uncertainty is considered as aleatory. I wonder how this is considered in the calculations. I assume this is done by some built-in procedure in the OpenQuake code itself. Is that right? A Montecarlo? If so, how this affect the 60 realizations? Could you provide some extra information about this?*

ANSW: Our text might induce some confusion in the reader. The aleatory uncertainty is recalled to indicate there is a probability function to describe the occurrences of the hypocentral depths. We modified the text to be clearer, as this is a consideration of how to treat the next occurrences, not an OpenQuake procedure.

*REF: - Rupture mechanism: In the same line as above (291) it is said that this issue is also considered aleatory. Please, provide some extra information so the reader can follow properly the way the uncertainty is taken into account and how it is eventually affecting the 60 final realizations. I wonder for example on how you consider the dip and strike of the ruptures in the calculations (line 300), are they horizontal or you are using some fixed values according to the rupture mechanism, for instance 30, 60 and 90 for reverse, normal and strike-slip? or is it a variable considered random?*

ANSW: as above, we modified the text to be clearer, as this is a consideration of how to treat the uncertainty, not an OpenQuake procedure. For many zones, strike/dip were evaluated by adopting the Pondelli et al (2020) procedure, but where this was not possible, we adopted fixed values that represent a balance between the need of exploring multiple orientation and computational machine time. Computed values were in the appendix, now we added also the "fixed" ones.

*REF: - GMPEs: To properly follow the results and discussion it is necessary to provide some extra information about the GMPEs used in the calculations, particularly the "distance parameter". I also missed some information about the significance of these GMPEs to be used in Italy, about the distance and magnitude range considered in them, the rupture mechanisms, number of records, ... and very importantly: for what type of ground are you using the GMPEs (I assume rock-type, but this should be stated in the paper to properly interpret the results). I know this paper is not about GMPEs but these are crucial information for understanding the results. I also wonder about the differences between the GMPE of Bindi et al 2014 and Bindi et al 2011. Are they derived from the same database?*

ANSW: Some extra information about the GMMs used in the calculations have been provided in the revised manuscript. In particular, a brief description of the procedure followed in MPS19 to select and weigh the GMMs will be added together with basic characteristics of the 3 models, such as distance metrics and calibration datasets (e.g., dataset including European and Middle-East records for Bindi et al. 2014 and Italian dataset for Bindi et al. 2011). We also specified that hazard computation in our study, as well as in MPS19, was performed for rock-site conditions, i.e. EC8 site category A or Vs,30>800m/s.

*REF: - Macroareas (line 380): Macroareas (a set of grouped source areas) are used to calculate b-values. This procedure is followed so the fitted b value results statistically stronger than the one doing the fitting in each of the zones. This a practical procedure, however it may miss significant b variations from zone to zone. It would be good to support the use of this "concept" a bit more.*

ANSW: We expanded this part to better explain why we used macroarea. In particular, our main concept is that for zones characterized by a "low" seismic activity, the b-value can be biased and results as an artefact of the low number of data available. Using macroarea, where zones were grouped according to their tectonic features, for example zones characterized by extension, we can estimate a statistically robust b-value. To maintain b variations from zone to zone we also used the approach 1, in which the b-value is actually estimated for each zone. Actually, we do not know what is the best solution: b-variable at the level of the single zone or at the level of large tectonic features. This is the reason why we used a logic tree approach.

*REF: - I suggest the authors to write at some point in the paper the "return periods" of the key annual probability of exceedance levels targeted (for example: 10% of exceedance in 50 years, also refer as 475-yr return period; and so on). This is not crucial, of course, but it helps the reader, particularly among the engineering community.*

ANSW: We agree with the Reviewer and we added the "return periods" of the key annual probability of exceedance levels targeted.

*REF: - **Discussion and Conclusion: I believe this section could be much improved.** I suggest you to separate Discussion from Conclusion. As it is written now, is seems a bit erratic. It is just a matter of organizing ideas and end properly with a short Conclusion.*

ANSW: We accepted this suggestion and we modified the section.

*REF: - line 356: please provide a bit more information on the "community-based effort". Was a procedure like SSHAC followed? Did it follow a sort of expert judgment method?*

ANSW: We added in the text the MPS19 project involved more than 150 Italian researchers at various stages of the project. Many of them were involved in the building of their seismicity model or earthquake rupture forecast. There were 11 groups of researchers that produced 11 ERFs, MA4 is one of them. Other researchers were involved in the selection of the GMMs and others on the testing procedure.

*REF: - line 363: refrain the use of "true tectonics", use instead "actual" or "known" for example.*

ANSW: We modified it according to the suggested term.

*REF: - line 364: Documentation is crucial in the process of defining source zones for PSHA. It supports the zone model and provides a ground for further refinements in future updates. The paper lists somehow the different data used in the process of defining the zones; however, it would be very good to provide detail information on each zone about the method/criteria used to define each of the boundaries (and may be add this info as an electronic supplement), as other authors have done elsewhere (eg., Vilanova et al., 2014; García-Mayordomo, 2015)*

ANSW: This is a good point, also raised by the REV1. We decided that all the data (georeferenced maps of the geophysical data and shapefiles) we used to draw the zones can be available by request. We agree that transparency is an important topic, but we would avoid copyrights issue. W also added details of group of zones, for example Alps, Northern Italy and so on.

*REF: - References:  There are few typos, eg., lines 478, 480,*

ANSW: we fixed these typos.

*REF: - Figure 6. I believe the y axis should read cumulative annual rate. Additionally, could you use a clearer scale for the x axis so it reads integers and halves (eg, 4.5, 5.0, 5.5,..). The graphs would look better if you also reduce a couple of marks the length of y-axis. In the caption, use approach i instead of "method 1".*

ANSW: yes, it is a cumulative annual rate. We modified the graphs according to these comments.

REF: - Figure 8. It would be good to also stated the "return period" of each p.o.e. NOTE there is a typo in the titles of the maps as it says 50 years twice, when it should read just "PGA 10% of p.o.e in 50 years". I assume is PGA on rock, but it would be good to say it.

ANSW: We added the RP and modified the graphs according to these comments. We corrected the typo.

*REF: - Figure 11: NOTE the typo on the right hand map (in the title). It should say 2%.*

ANSW: we fixed this typo.

*REF: - Figure 13: Typo in the caption, it says "againt".*

ANSW: we fixed this typo.

Dear Referee 3,

Thank you for your useful suggestions and comments. We appreciated the detailed review and we modified the text according to your comments.

In the following we answer point by point to your comments.

*REF: The manuscript is well organized, clearly written, the results are sound and supported by the data, the references are adequate and most figures are clear. However, the manuscript is rather technical and relatively short, allowing only a superficial analysis. In addition, there are several issues that need to be improved or clarified:*

*At some points, the text is too concise to fully understand, for example:*

*§2.4 "We also considered the regional strain rate fields … and the … Shmax orientation to qualitatively check the homogeneity of the strain rate values within the area sources": it is not clear how this is done, perhaps it could be shown in a map in the electronic supplement.*

ANSW: We agree with the Reviewer that this part should be better discussed. We decided that all the data (georeferenced maps of the geophysical data and shapefiles) we used to draw the zones can be available by request. We agree that transparency is an important topic, but we would avoid copyrights issue. In these files, we added details of group of zones, for example Alps, Northern Italy and so on.

*REF: §3.1, e) "be consistent with the CPTI15 earthquake catalogue": how specifically?*

ANSW: We referred to the spatial pattern of seismicity. We would be sure that zones whose borders were drawn using faults were not across a group of earthquakes attributed to the same faults. The non-precise correspondence between faults and earthquake is due to the fact that earthquakes occurred before the 1950 are parametrized using macroseismic data, then a shift between the possible causative faults and the epicentre can occur.

*REF: §3.2 "We considered only the earthquakes that can be related to active crustal seismicity based on the crustal models by …": it is not clear which (types of) events are rejected*

ANSW: We specified better in the text that the we used crustal model to identify earthquakes whose depth is shallower than the moho.

*REF: §3.3 "in each source zone we obtained a representative moment tensor": how was this done?*

ANSW: We agree and we furnish some detail on the procedure, instead simply refer to Pondrelli et al (2020).

*REF: The description of methods iv and v to calculate seismic rates in §3.4.3 is too concise.*

ANSW: We detailed the description of these approaches.

*REF: The introduction is actually a summary of the various seismic hazard models in Italy and their evolution during the past 2 decades. Considering the numerous abbreviations, a sketch depicting the model hierarchies and histories would be useful.*

ANSW: We thank the reviewer for this suggestion. We included a sketch in Figure 1cto depict hierarchies and histories of ZS4,ZS9, ZS16, MPS04, MA4 and MPS19.

*REF: Declustering of earthquake catalogs is an important issue in modern, state-of-the-art probabilistic seismic hazard assessments, but is not really investigated in this study. Although many different methods are available, only a single one is used. It is not clear to me why this would be less important than for instance the different methods to estimate the completeness. The latter are captured in a branching level of the logic tree, but declustering is not.*

ANSW: The earthquake catalogue used to evaluate seismic rates was furnished by the Italian project of MPS19. We agree that also the declustering approach should be part of a logic tree exploration, however, this was not done in the framework of the MPS19 project. We added this point in the section on earthquake catalogues.

*REF: The logic-tree structure used to calculate the ERFs is described at the beginning of §4 (seismic hazard calculation), but I would prefer to move this to §3.4, as it provides the rationale for the choices made in the subsections of §3.4.*

ANSW: We modified the introduction of the Section 4 according to this suggestion.

*REF: Figures 9 and 10 show hazard results for 3 selected sites, but there is no discussion about what we can learn from the differences between these sites.*

ANSW: The 3 sites were chosen only as examples for relatively low-to-high seismic hazard levels in Italy.

*REF: I have not been able to understand the analysis of ERF and GMM uncertainties at the end of §4 and in Fig. 13. According to the text, the curves grouped by GMM are used to analyse the ERF uncertainty, whereas I would think that these curves show the differences in uncertainty among the different GMMs (e.g., there is a clear divergence for the Bindi et al (2012) GMM at low APOE in L'Aquila and Siracusa). Furthermore, it is not clear to me how we should compare the other set of curves (with marker symbols) with the ones grouped by GMM, considering that the former represent 3 branches and the latter 20 branches. It may be a misunderstanding on my part, but I think it would be more useful to also group these curves by completeness model, by Mwmax model and by AR model. This would also reduce the number of curves in Fig. 13. If this is not what the authors intend to show, then a more clear description will be needed to understand the conclusion that "there is a clear trend that ERF uncertainty gives larger CoV than GMM uncertainty". Figure 13 is too dense: it shows 24 curves with different colors, line styles and marker symbols, but many of them cannot be properly distinguished.*

ANSW: We simplified these figures according to the suggested procedure. We think this will help to better visualise and describe the uncertainty due to ERF and GMM.

*REF: There is no caption for the tables in the electronic supplement.*

ANSW: We included captions.

*REF: Detailed comments. I have annotated a number of minor comments, corrections and suggestions in the attached PDF. The easiest way to view these is by opening the Comment side panel in Adobe Acrobat Reader.*

ANSW: We opened the pdf. We thank the Reviewer for the detailed comments that we have taken into consideration. We modified the text according to the suggested comments.